# Black phosphorene as a hole extraction layer boosting solar water splitting of oxygen evolution catalysts

Kan Zhang [1,2], Bingjun Jin [2], Cheolwoo Park[3], Yoonjun Cho[2], Xiufeng Song [1], Xinjian Shi [4], Shengli Zhang[1], Wooyul Kim[3], Haibo Zeng [1] & Jong Hyeok Park[2]

As the development of oxygen evolution co-catalysts (OECs) is being actively undertaken, the tailored integration of those OECs with photoanodes is expected to be a plausible avenue for achieving highly efficient solar-assisted water splitting. Here, we demonstrate that a black phosphorene (BP) layer, inserted between the OEC and $BiVO_4$ can improve the photoelectrochemical performance of pre-optimized $OEC/BiVO_4$ (OEC: NiOOH, $MnO_x$, and CoOOH) systems by 1.2~1.6-fold, while the OEC overlayer, in turn, can suppress BP self-oxidation to achieve a high durability. A photocurrent density of $4.48\,mA\cdot cm^{-2}$ at 1.23 V vs reversible hydrogen electrode (RHE) is achieved by the $NiOOH/BP/BiVO_4$ photoanode. It is found that the intrinsic $p$-type BP can boost hole extraction from $BiVO_4$ and prolong holes trapping lifetime on $BiVO_4$ surface. This work sheds light on the design of BP-based devices for application in solar to fuel conversion, and also suggests a promising nexus between semiconductor and electrocatalyst.

---

[1] MIIT Key Laboratory of Advanced Display Material and Devices, School of Materials Science and Engineering, Nanjing University of Science and Technology, Nanjing 210094, China. [2] Department of Chemical and Biomolecular Engineering, Yonsei University, 50 Yonsei-ro, Seodaemun-gu, Seoul 120-749, Republic of Korea. [3] Department of Chemical and Biological Engineering, Sookmyung Women's University, Seoul 04310, Republic of Korea. [4] Department of Mechanical Engineering, Stanford University, Stanford, CA 94305, USA. Correspondence and requests for materials should be addressed to K.Z. (email: zhangkan@njust.edu.cn) or to H.Z. (email: zeng.haibo@njust.edu.cn) or to J.H.P. (email: lutts@yonsei.ac.kr)

Photoelectrochemical (PEC) water splitting on polycrystalline BiVO4 photoanodes has attracted considerable attention in recent years due to the narrow bandgap (2.4–2.5 eV) and deep valence band edge of BiVO4, which enable visible light harvesting and water oxidation[1,2]. However, the occurrence of surface/bulk charge recombination due to the poor charge transport characteristics and short hole-diffusion length (<70 nm) of BiVO4 leaves room to improve the PEC performance of BiVO4 photoanodes[3,4]. Heteroatom doping[5–7], component or structural tuning[8–10], and loading of oxygen evolution co-catalysts (OECs)[11–14] are identified as the most promising approaches for overcoming these drawbacks and improving the PEC performance of BiVO4 photoanodes. Among these methods, OEC loading can strongly suppress surface recombination in BiVO4 photoanodes and also shift the photocurrent onset potential close to its flat-band potential for water oxidation, which is the most significant feature for achieving unbiased solar water splitting[15,16].

Recently, van de Krol et al. re-stated the roles of some OECs, such as Co-Pi and RuOx, from the perspectives of the surface reaction kinetics and surface recombination. The researchers mainly pointed out that the water oxidation capability of OECs is strongly limited by their small thermodynamic driving force caused by insufficient hole extraction from the photoanodes[17]. This limitation means that for one of the best existing photoanode materials with an OEC, BiVO4/OEC, the band bending of BiVO4 at the electrode/electrolyte interface must be optimized[18–20]. For instance, Kim and Choi demonstrated that the incorporation of a FeOOH compound can accelerate hole transport from BiVO4 to the NiOOH OEC because the hole transport resistance of FeOOH is lower than that of NiOOH[13]. Zhong et al. suggested that the deposition of p-NiO on the CoOx OEC/BiVO4 surface to form a p–n junction interface can be beneficial for rapid hole extraction to reduce bulk charge recombination in BiVO4[21]. Gong's group directly employed a p-Co3O4 OEC instead of CoOx and proved that a p-type semiconductor having OEC functions can result in simultaneous enhancements in the hole extraction and water oxidation capabilities of the BiVO4 photoanode[22]. Therefore, promoting hole extraction from BiVO4 to OECs by improving their interface resistance still holds broad interest and significance for enhancing the PEC performance.

As a novel 2D family of materials, exfoliated black phosphorene (BP) layers that are 2–20 nm thick can show p-type semiconductor properties with high hole mobility (1000 $cm^2 V^{-3}s$), which are caused by the unavoidable presence of oxygen species[23,24]. On the other hand, its bandgap properties, which are dependent on the number of layers, result in a tunable bandgap between the bulk value of 0.3 eV to the monolayer value of 2.1 eV; therefore, BP is considered a photoabsorber of visible and near-infrared solar light for solar light harvesting[25]. Apart from several compelling succeeds in the application of exfoliated BP as a photocatalyst for $H_2$ generation and water splitting[26–30], employing exfoliated BP and its tailored integration with photoanodes to enhance the PEC performance for highly efficient water splitting has not been given much attention[31].

In this study, we first demonstrate that the insertion of exfoliated BP nanosheets with ~4 layers between BiVO4 photoanodes and conventional OEC layers can lead to ultra-rapid hole extraction. The electrochemical analysis reveals a built-in p/n electric field formed by the BP/BiVO4 heterostructure, in which the space-charge region results in an upward shift in the energy level of the BP nanosheets. After coating of the photoanode with an additional thin OEC layer (NiOOH), the interfacial band-edge energetics strongly drive holes from BiVO4 to the NiOOH surface for efficient water oxidation. As a result, NiOOH/BP/BiVO4 achieves a photocurrent density of 4.48 mA·$cm^{-2}$ at a bias of 1.23 V vs. RHE, which is 4.2 times higher than that of pure BiVO4 and 1.5 times higher than that of NiOOH/BiVO4. Moreover, the hole extraction role of the BP nanosheets is successfully evidenced by two other OECs (MnOx and CoOOH), demonstrating the potential of BP as an auxiliary to enhance water oxidation.

## Results

**Characterization of the BP/BiVO4 photoanode**. BP nanosheets were synthesized by liquid exfoliation of bulk BP particles and dispersed in isopropanol (IPA) under an $N_2$ atmosphere (Supplementary Fig. 1). The redshifted Raman signals of the BP nanosheets confirm the successful exfoliation of bulk BP (Supplementary Fig. 2). The atomic force microscopy (AFM) image of the exfoliated BP nanosheet layers shows a distinct 2D morphology with an average thickness of ~2.2 nm, corresponding to 4 layers (Supplementary Fig. 3)[32]. High-resolution transmission electron microscopy (HR-TEM) images of the exfoliated BP nanosheets display clear lattice fringes with a d-spacing of 0.34 nm, corresponding to the (040) plane (Supplementary Fig. 4). A nanoporous BiVO4 photoanode was fabricated by using an electro-deposited BiOI film as a precursor based on the previous method[13]. The thickness of the as-prepared BiVO4 photoanode was ca. 1 µm (Fig. 1a). Considering that the lateral size of BP is larger than the pore size of BiVO4 film, the depositing BP on BiVO4 photoanode was assisted by centrifuge-coated method (See experimental section for detail). Compared to the morphology of the pure BiVO4 photoanode (Fig. 1b and Supplementary Fig. 5a), the SEM image of BP/BiVO4 does not reveal the presence of BP nanosheets on surface of BiVO4 photoanode (Fig. 1c and Supplementary Fig. 5b), which is in stark contrast to that the BiVO4 photoanode is immersed into the BP dispersion by natural adsorption or deposition (Supplementary Fig. 6). X-ray diffraction (XRD) analysis demonstrates monoclinic BiVO4 crystal, which remains unchanged after the deposition of BP, but a small diffraction peak of BP can be detected (Supplementary Fig. 7). However, although the observation on BP/BiVO4 by TEM image could not distinguish the presence of BP nanosheets (Fig. 1d), the electron diffraction spot confirms the co-existence of polycrystalline BiVO4 and BP components (insert in Fig. 1d). High-angle annular dark-field scanning TEM-energy-dispersive spectroscopy (HAADF-STEM-EDX) reveals an obvious sheet-like distribution pattern of P element clinging to BiVO4 particles (Fig. 1e). Further enlarged HAADF-STEM image exhibits a bright area on the BiVO4 particle, which can be identified as a BP sheet (Supplementary Fig. 8). However, the structure incompatibility between the 2D BP sheets and 3D BiVO4 nanopores may lead to uncovered part of BiVO4 by BP. Nevertheless, its high-resolution TEM (HR-TEM) image shows the distinct interface of BP/BiVO4, in which the lattice spacing of 0.212 nm corresponds to the (051) planes of monoclinic BiVO4[14], while the other lattice spacing of 0.54 nm is consistent with the interlayer distance of BP along the c-axis[33]. Compared to the BP/BiVO4 prepared by the centrifuge-coated method, the naturally deposited BP/BiVO4 shows the poor connection between the BP sheet and BiVO4 particles (Supplementary Fig. 9). The conductivity of the BiVO4 photoanode and BP nanosheets was investigated with Mott-Schottky plots. As shown in Fig. 1g, BiVO4 is a typical n-type semiconductor with a Fermi energy of 0.324 V vs NHE; the n-type behaviour is usually caused by the presence of oxygen defects[1]. In contrast, the straight line of exfoliated BP displays a negative slope with a Fermi level of 0.588 V vs NHE, indicating p-type conductivity. The intimate contact between BiVO4 and the BP nanosheets makes it easy to construct an electric field from the p/n junctions. High-resolution X-ray photoelectron spectroscopy (XPS) of Bi, O, and V can determine the built-in potential and band offsets at the

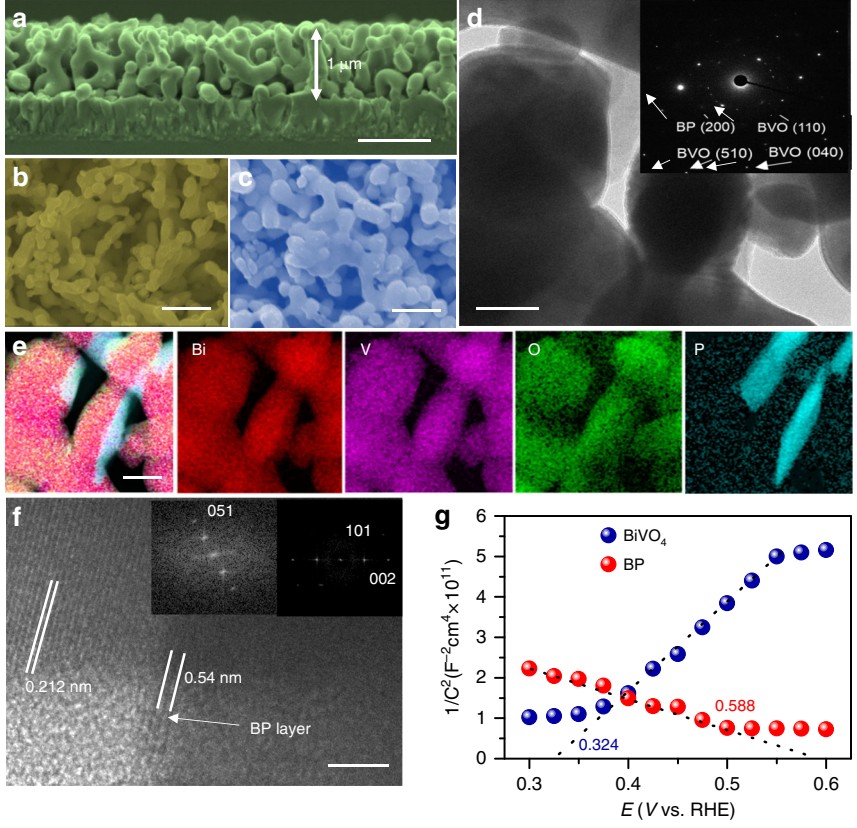

**Fig. 1** Morphology and conductivity characterizations. **a** Cross-section SEM image, scan bar: 1 μm. **b** Top view SEM images of as-prepared BiVO$_4$ photoanode and (**c**) the top view SEM image of BP/BiVO$_4$ photoanode, scan bar: 500 nm. **d** TEM image of BP/BiVO$_4$ photoanode, scan bar: 100 nm, insert is electron diffraction spot. **e** HAADF-STEM-EDX element mapping and **f** HR-TEM image of BP/BiVO$_4$ heterojunction, scan bar: 5 nm. **g** Mott-Schottky plots of BP and BiVO$_4$ electrodes measured with a frequency of 500 Hz and amplitude of 10 mV in KPi electrolyte (pH = 7.1)

interface[34]. As shown in the Bi 4f (Fig. 2a) and O 1s/V 2p (Fig. 2b) XPS spectra, the Bi 4f$_{7/2}$ core-level and O 2p core-level spectra of BP/BiVO$_4$ shift by 0.3 and 1.58 eV, respectively, to higher binding energies in comparison to those of pure BiVO$_4$, whereas the V 2p$_{3/2}$ core-level spectra of BP/BiVO$_4$ and BiVO$_4$ are almost the same. Accordingly, the binding energy of the P 2p core-level spectrum of BP shifts by ~0.41 eV to lower energy after integration with BiVO$_4$ (Fig. 2c). The differential charge density diagram of the BP/BiVO$_4$ heterointerface was compared with that of the clean BiVO$_4$ surface by Bader charge analysis, which can further reveal the charge transfer direction (Fig. 2d). In detail, BP with a 1 × 3 supercell donates 0.12 e to BiVO$_4$ with a 1 × 2 supercell, and the transferred charges are mainly distributed on the interfacial O atoms with negligible effect on the V atoms. Density of states (DOS) calculations were then conducted to determine the electronic structure of BiVO$_4$. The calculated valance band (VB) maximum and conduction band (CB) minimum of monoclinic BiVO$_4$ with a standard space group of $C2/c$ is mainly comprised of O 2p and V 3d orbitals (Supplementary Fig. 10), which is in good agreement with a previous report[35]. Remarkably, BP profoundly influences the VB electronic structure of BiVO$_4$ through the overlap of the P 2p and O 2p orbitals independent of the CB minimum of BiVO$_4$ (Fig. 2e). As the O 2p and V 3d orbitals contribute to the VB of BiVO$_4$, the charge transfer that occurs on the O and V atoms at the $p/n$ junction might be implicated in upward bending of the VB. As shown in Fig. 2f, The VB position is shifted upward from 2.48 eV for BiVO$_4$ to 2.23 eV for BP/BiVO$_4$, implying that the positrons (holes) are the dominant carriers across the $p/n$ junction under reverse bias[36]. According to the above experimental and theoretical

results, Fig. 2g summarizes the possible band offsets and built-in potential of the BP/BiVO$_4$ heterojunction. Since the BP/BiVO$_4$ photoanode performs under external bias to facilitate electron transport from BP/BiVO$_4$ to the counter electrode in PEC water splitting, the external bias is therefore regarded as reverse bias. Positrons as the dominant carriers across the BP/BiVO$_4$ heterointerface would promote hole extraction from BiVO$_4$ to BP under external bias. To determine the hole extraction as a function of applied bias, the in-situ ultrafast transient absorption (TA) spectroscopy is performed to evaluate the hole trapping behaviours. As reported previously, the TA signal of surface-trapped holes for BP/BiVO$_4$ hybrid is mainly located in the wavelength ranging from 400 to 700 nm[27]. Therefore, the TA signal at 500 nm for BiVO$_4$ and BP/BiVO$_4$ anodes, respectively under open circuit potential and 0.8 V vs. Ag/AgCl in 0.5 M phosphate buffer (KPi, pH 7.1) are monitored. Compared with BiVO$_4$ anode, the intensity of the absorption signal for BP/BiVO$_4$ anode is significantly increased as anodic shifting the applied bias (Supplementary Fig. 11). In Fig. 3a, b, the decay signal was further fitted to a biexponential decay model with a fast component ($\tau_1$) and a slow component ($\tau_2$)[37,38], and the lifetimes are summarized in Fig. 3c. Since the fast component, $\tau_1$, is associated with the hole being trapped at near band edge, the decreased $\tau_1$ values for both BiVO$_4$ and BP/BiVO$_4$ photoanodes under applied bias can be considered the external bias that boosts hole transport[37,38]. The BiVO$_4$ photoanode shows similar $\tau_2$ value with 141.97 μs, 14.5% under OCP and 141.17 μs, 27.8% at 0.8 V bias, whereas the BP/BiVO$_4$ anode exhibits an approximate two-fold increase in $\tau_2$ value from 147.15, 13.9% under OCP to 280.25 μs, 14.4% at 0.8 V bias. The slow component, $\tau_2$, can be ascribed to the holes being

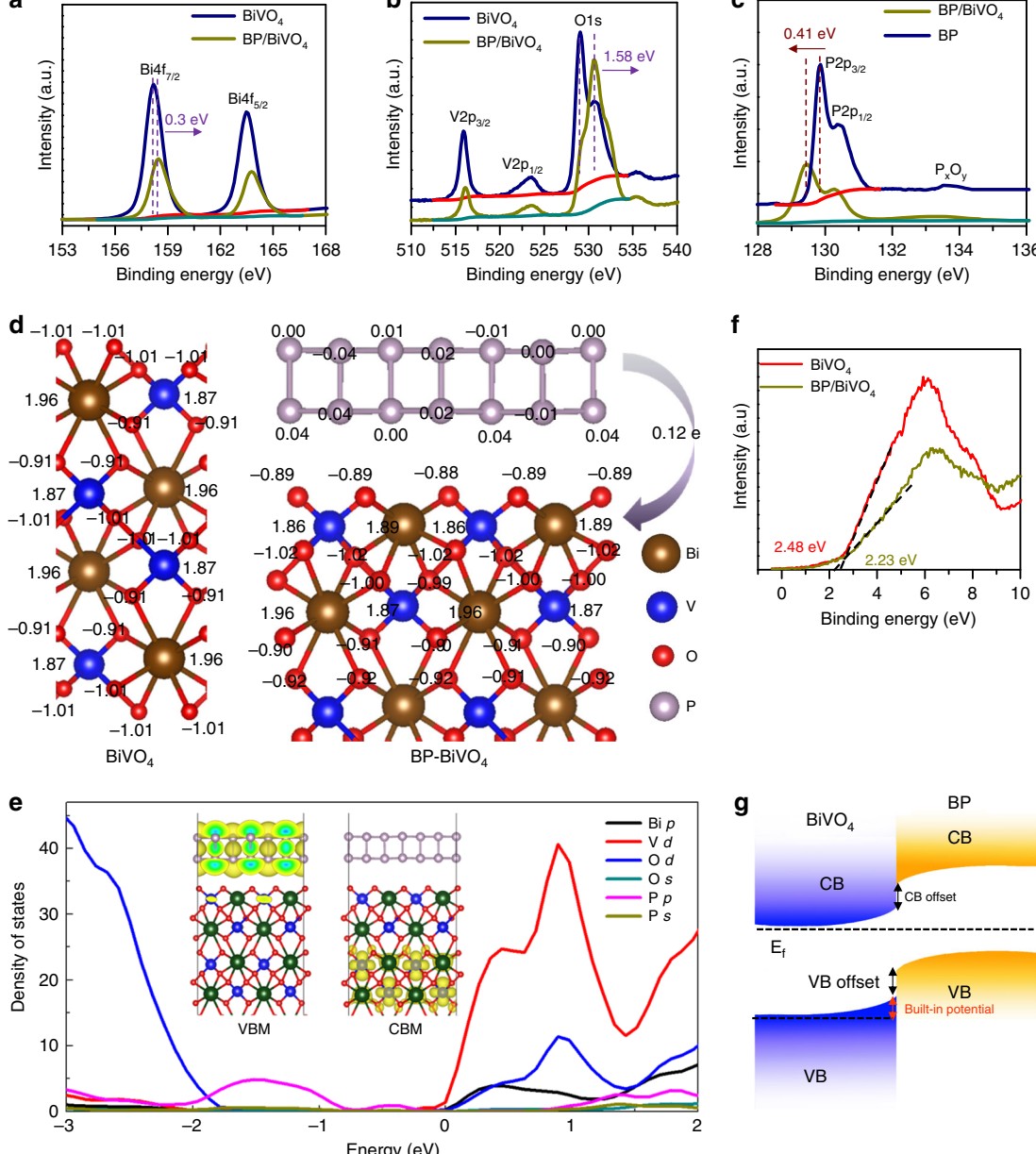

**Fig. 2** Electronic structure characterizations. **a** Bi 4*f*, (**b**) V 2*p* and O 1*s* XPS of BiVO₄ and BP/BiVO₄ photoanodes, (**c**) P 2*p* XPS of BP and BP/BiVO₄ photoanode, **d** The charge density difference between BiVO₄ and BP/BiVO₄, (**e**) VB XPS of BiVO₄ and BP/BiVO₄ photoanodes, (**f**) DOS of BP/BiVO₄ heterojunction, (**g**) Energy diagram of BP/BiVO₄ heterojunction interface

trapped at photoanode/electrolyte interface, which is expected to yield long-lived holes for water oxidation[37]. For BP/BiVO₄ anode, the two-fold increase in slow component suggests that the increased number of holes are extracted from BiVO₄ to photo-anode/electrolyte interface under applied bias, which is consistent with our assumption from their band alignment.

**PEC Performance Evaluation of BP/BiVO₄ anode**. The improvement in the PEC performance of BiVO₄ imparted by the *p*–*n* junction formed with BP was investigated by measuring the *J*–*V* curves in 0.5 M phosphate buffer (KPi, pH 7.1) from rear side illumination (AM 1.5G, 100 mW cm⁻²). As shown in Fig. 3d, an obvious improvement in the photocurrent density of BiVO₄ is observed in the presence of BP, whereas the dark current density demonstrates that the water oxidation kinetics are slower for

BP/BiVO₄ than for BiVO₄ (Supplementary Fig. 12). These results imply that the enhanced photocurrent density of BP/BiVO₄ does not originate from surface water oxidation. As a consequence, a NiOOH layer that is a well-defined OEC for water oxidation was electro-deposited on the BP/BiVO₄ electrode to enhance the water oxidation kinetics (Supplementary Fig. 13)[39]. SEM and TEM images of NiOOH/BP/BiVO₄ reveal a near-surface BP nanosheet layer buried by an amorphous NiOOH layer (Supplementary Fig. 14). Remarkably, the NiOOH/BP/BiVO₄ photo-anode achieves a photocurrent density of 4.48 mA cm⁻² at 1.23 V vs NHE, which is 1.5-fold higher than that of the NiOOH/BiVO₄ photoanode (3.03 mA·cm⁻² at 1.23 V vs. NHE) and 2.7-fold higher than that of the BP/BiVO₄ photoanode (1.66 mA cm⁻² at 1.23 V vs NHE). As reported by Kim and Choi[13], the interfacial resistance between BiVO₄ and NiOOH creates an energy barrier that impedes rapid hole transfer to the reaction surface, so the

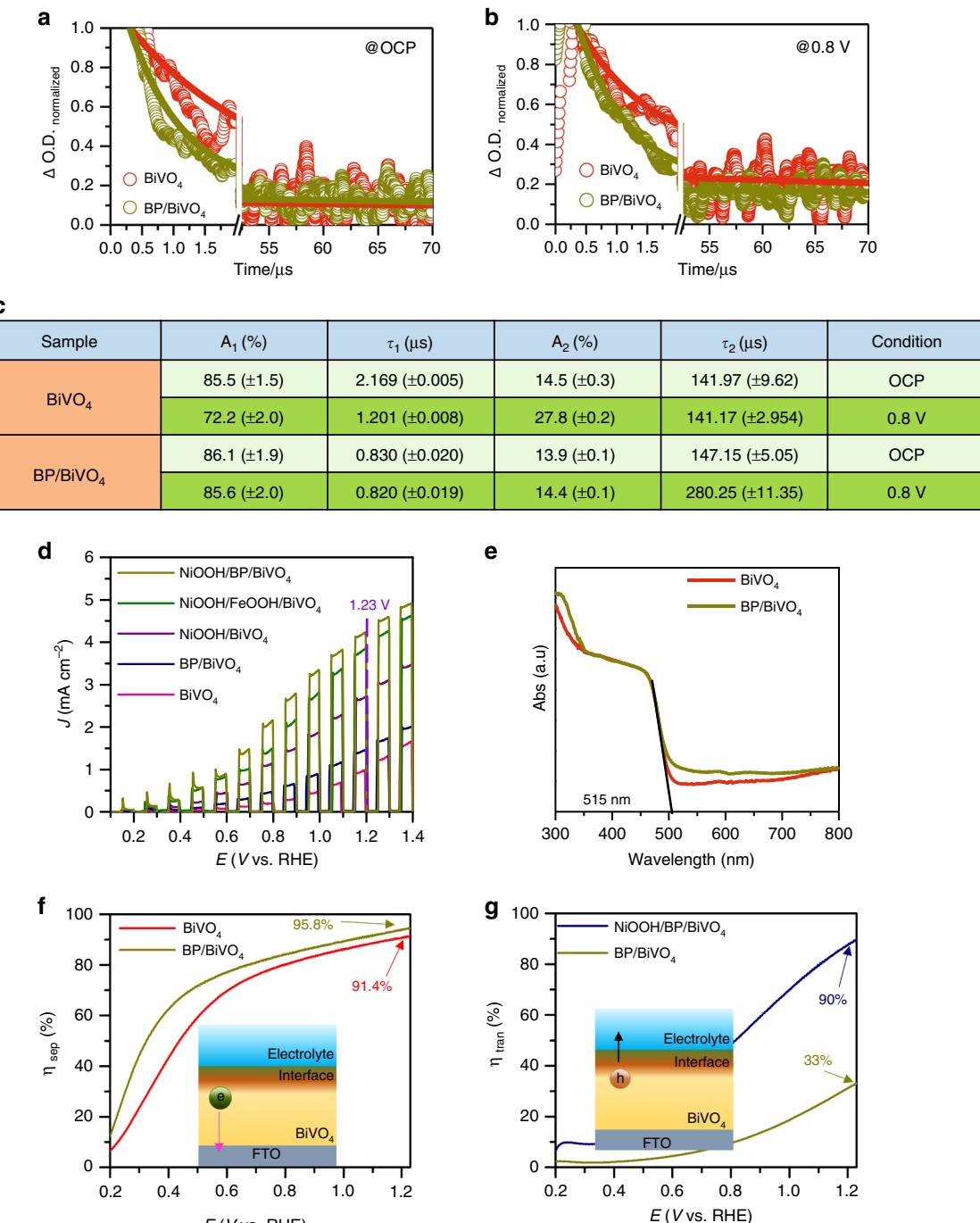

**Fig. 3** In-situ TA spectroscopic evidence for hole extraction as a function of applied bias. The decay recorded at 500 nm under OCP and 0.8 V vs Ag/AgCl in 0.5 M KPi electrolyte (pH = 7.1) was fitted to a biexponential decay model, $y = y_0 + A_1 e^{-(x-x_0)/t_1} + A_2 e^{-(x-x_0)/t_2}$ for (**a**) BiVO$_4$ and (**b**) BP/BiVO$_4$. **c** Fitting parameters of the TA signals. PEC performance, optical properties and electrochemical characterizations. **d** Chopped J–V curves of various photoanodes in KPi electrolyte (pH = 7.1) under AM 1.5 illumination. **e** UV–Vis absorbance of BiVO$_4$ and BP/BiVO$_4$ photoanodes. **f** Charge separation efficiencies of BiVO$_4$ and BP/BiVO$_4$ photoanodes. **g** Charge transfer efficiencies of BP/BiVO$_4$ and NiOOH/BP/BiVO$_4$ photoanodes

insertion of a FeOOH layer between NiOOH and BiVO$_4$ facilitates hole transfer, hence enhancing the PEC performance. In our case, the NiOOH/FeOOH/BiVO$_4$ photoanode exhibits a photocurrent density of 4.13 mA·cm$^{-2}$ at 1.23 V vs NHE, which is lower than the photocurrent density of NiOOH/BP/BiVO$_4$. In contrast, the FeOOH/BiVO$_4$ photoanode exhibits a higher photocurrent density than BP/BiVO$_4$ (Supplementary Fig. 15). These results clearly illustrate that the superior PEC performance of the NiOOH/BP/BiVO$_4$ photoanode relative to that of NiOOH/

FeOOH/BiVO$_4$ results from the interfacial behaviour associated with hole transfer. To understand how hole transfer was improved by the presence of BP nanosheets, the charge separation efficiency ($\eta_{sep}$) was calculated by the following equation[40]:

$$\eta_{sep} = J_{HS}/J_{abs}, \quad (1)$$

where $J_{HS}$ is the photocurrent density measured in a hole scavenger-containing electrolyte and $J_{abs}$ is associated with the maximum photocurrent density ($J_{max}$) and light harvesting

efficiency (LHE). The UV–Vis absorption spectra of $BiVO_4$ and $BP/BiVO_4$ are shown in Fig. 3e. The spectrum of $BP/BiVO_4$ contains the unchanged absorption edge of $BiVO_4$ at 515 nm in addition to a longer absorption tail up to 800 nm. The expanded absorption region of $BP/BiVO_4$ can be ascribed to the narrower bandgap of BP (Supplementary Fig. 16). However, the BP nanosheets might not act as an efficient photosensitizer to inject electrons, as the $BP/BiVO_4$ photoanode does not show a detectable photocurrent response at 520 nm in the presence of $Na_2SO_3$ as a hole scavenger (Supplementary Fig. 17). As a result, $J_{max}$ and LHE can be established based on only the absorption of $BiVO_4$ ranging from 300 to 515 nm. The LHE and calculated $J_{max}$ value are shown in Supplementary Fig. 18 and the $J_{HS}$ is measured in the presence of $Na_2SO_3$ as a hole scavenger (Supplementary Fig. 19). The corresponding $\eta_{sep}$ value is shown in Fig. 3f. The BP nanosheets clearly significantly improve the $\eta_{sep}$ of $BiVO_4$ in the entire voltage region. Excluding the possibility of electron injection by the excited BP layer, the enhanced $\eta_{sep}$ directly points to efficient hole extraction from the excited $BiVO_4$ photoanode to the BP layer. Nevertheless, due to its poor water oxidation ability (Supplementary Fig. 12), the charge transfer efficiency ($\eta_{tran}$) of the $BP/BiVO_4$ photoanode, which is calculated as $J_{Ph}/J_{HS}$ ($J_{Ph}$ is the photocurrent density measured in KPi electrolyte)[41], is lower than that of the $BiVO_4$ photoanode (Supplementary Fig. 20). To understand the charge separation and transfer limitation, the $J$–$V$ curves of $BiVO_4$ and $BP/BiVO_4$ photoanodes were measured from front illumination. As shown in Supplementary Fig. 21a, the photocurrent densities of $BP/BiVO_4$ photoanode are still better than that of $BiVO_4$ in the KPi electrolyte with and without hole scavenger. The calculated $\eta_{sep}$ is similar to the result obtained by rear illumination (Supplementary Fig. 21b), whereas the calculated $\eta_{tran}$ (Supplementary Fig. 21c) display different tendency from rear illumination. The enhanced $\eta_{tran}$ of $BP/BiVO_4$ photoanode from front illumination is unexpected, which might be ascribed to the effect of surface passivation by BP layers on the reduction of surface recombination that compensates poor water oxidation ability[42]. Therefore, the subsequent introduction of the NiOOH overlayer on $BP/BiVO_4$ is crucial for improving the hole capability towards efficient water oxidation. $\eta_{tran}$ of $NiOOH/BP/BiVO_4$ (90%) is ~2.7 times higher than that of $BP/BiVO_4$, which has a value of 1.23 V vs RHE (Supplementary Fig. 22 and Fig. 3g), indicating the occurrence of strong synergistic effects between NiOOH and BP.

**Hole Extraction Behaviour of BP Nanosheets**. The capability of BP to perform hole extraction is further proven by electrochemical impedance spectroscopy (EIS) conducted at 0.6 V vs RHE in KPi electrolyte (Fig. 4a). The Nyquist plots were fitted by an equivalent circuit, as shown in the inset of Fig. 4a and the results are displayed in Supplementary Table 1. The NiOOH overlayer clearly significantly reduces the charge transfer resistance (R2) of both $FeOOH/BiVO_4$ and $BP/BiVO_4$ due to the enhanced water oxidation capability, while $NiOOH/BP/BiVO_4$ possesses a smaller charge transport resistance (R2) than $NiOOH/FeOOH/BiVO_4$[41]. Based on the smaller charge transport resistance of $BP/BiVO_4$ relative to that of $BiVO_4$, the results clearly show that the hole extraction behaviour of the BP nanosheets is superior to that of FeOOH. Furthermore, the bulk capacitances ($C_{bulk}$) for all photoanodes are almost same, which can be ascribed to the redox process of $V^{4+}/V^{5+}$ [19,41]. The capacitances at the electrode/electrolyte interface for $BP/BiVO_4$, $NiOOH/FeOOH/BiVO_4$ and $NiOOH/BP/BiVO_4$ are significantly increased, which can be related to the surface layer that modifies surface state of $BiVO_4$[43]. Figure 4b demonstrates the occurrence of long-lived hole storage by the BP nanosheet layer based on

analysis of the transient cathodic current. $t_1$, corresponding to photocurrent quenching under instantaneous light-off conditions, gradually decays to a steady state ($t_2$), thus showing a cathodic current. The delay in the steady-state cathodic current indicates that the separated holes that reach the electrode/electrolyte interface are not involved in water oxidation but are instead stored at the electrode surface. Therefore, the large value of $t_2$-$t_1$ for $BP/BiVO_4$ indicates the presence of long-lived holes at the surface of $BiVO_4$[44–46]. The charge storage behaviour of the $NiOOH/BP/BiVO_4$ photoanode against an applied bias can be calculated from the transient-state photocurrent based on the chronoamperometry curve measured under chopped illumination and linear sweep voltammetry (LSV) curves, respectively (Supplementary Fig. 23). The photocurrent drop from the transient state to the steady state can be ascribed to the number of holes stored[43]. Compared to the $NiOOH/BiVO_4$ photoanode, the number of holes stored by the $NiOOH/BP/BiVO_4$ photoanode is obviously higher across the entire potential region, especially at low bias (Fig. 4c). The fate of the holes extracted is to reach the surface, then participates in the water oxidation reaction at high potentials, where the injection barrier no longer impedes the charge transfer from the electrode to the electrolyte. These results further demonstrate the strong capability of the BP nanosheet layer to perform hole extraction towards water oxidation occurring at surface of OECs.

To further illustrate the impressive role of the BP nanosheet layer in hole extraction, two other well-defined OECs, CoOOH, and $MnO_x$, were respectively spin-coated and photo-deposited on the $BP/BiVO_4$ photoanode surface (Supplementary Fig. 24). The enhancement in the PEC performance is evidenced by comparison of the catalysts deposited on $BiVO_4$ photoanodes, as demonstrated by the cyclic voltammetry (CV) curves in Fig. 4d. The enhancement factors induced by the BP layer are summarized in Supplementary Table 2, in which an average 1.5-fold enhancement is observed, and the NiOOH OEC overlayer demonstrates the highest PEC performance, which arises from its superior water oxidation capability (Supplementary Fig. 25).

**PEC Water Splitting of NiOOH/BP/BiVO₄ anode**. The BP nanosheet layer buried underneath the NiOOH layer exhibits a current density of 4.46 mA cm$^{-2}$ at 1.23 V vs NHE for at least 200 min, as shown in Fig. 5a. Without the outmost NiOOH layer or with pure $BiVO_4$ ($BP/BiVO_4$ or $BiVO_4$ photoanode), the steady-state current density at 1.23 V vs NHE gradually drops. The fading of the photocurrent density can be ascribed to anodic photocorrosion of $BiVO_4$ by the surface-accumulated holes that arise from an insufficient water oxidation capability[47,48]. In addition, the oxygen gas surrounding BP may cause its self-oxidation, as determined by P 2$p$ XPS analysis (Supplementary Fig. 26). Moreover, $BiVO_4$ combined with BP and NiOOH layers demonstrates superior PEC performance relative to other OEC/$BiVO_4$ photoanodes and competitive values with those improved OEC/$BiVO_4$ photoanodes (Supplementary Table 3). The gas evolved from the $NiOOH/BP/BiVO_4$ photoanode was measured from the photocurrent density at 1.23 V vs RHE. As shown in Fig. 5b, the linear fitting plots of both $H_2$ and $O_2$ nearly overlap with the theoretical number of electrons. The excellent Faradic efficiency for $O_2$ evolution indicates that the hole-storing behaviour of BP does not impede the oxygen evolution reaction taking place at the NiOOH surface.

Furthermore, the long-term durability of the $NiOOH/BP/BiVO_4$ photoanode was investigated at 1.23 V vs RHE. This test was performed for 60 h and rested for 2 h with an interval of 20 h. As shown in Fig. 5c, the photocurrent density of the

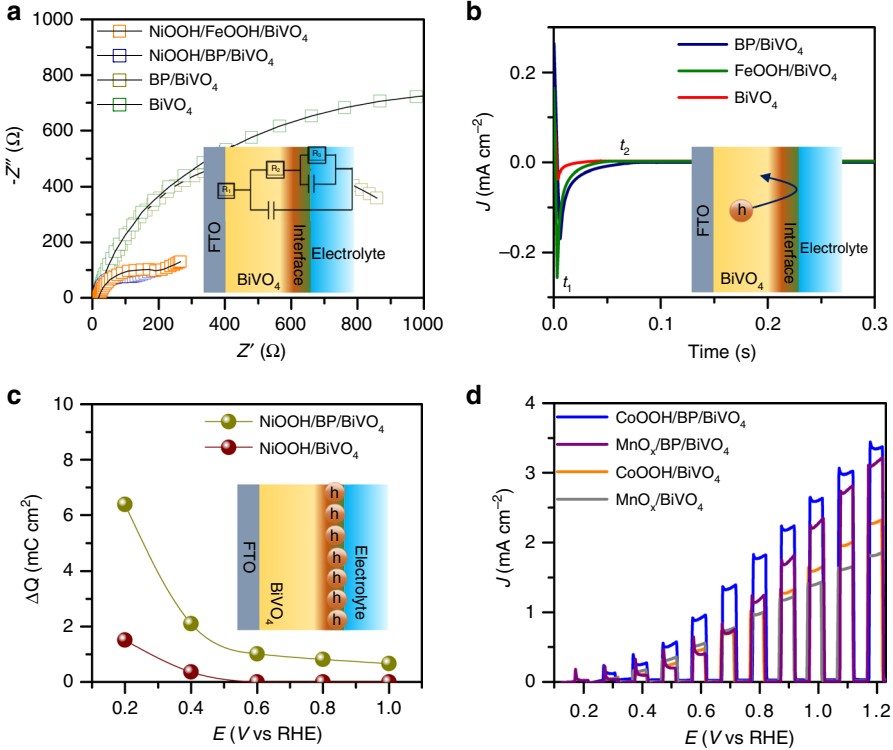

**Fig. 4** PEC performance and electrochemical characterizations. **a** EIS Nyquist plots measured at 0.6 V vs RHE in KPi electrolyte (pH = 7.1) under AM 1.5 illumination, **b** Delay of the cathodic photocurrent curves measured at 0.2 V vs RHE in KPi electrolyte (pH = 7.1) under AM 1.5 illumination, **c** Charge storage capability against applied bias for both NiOOH/BiVO$_4$ and NiOOH/BP/BiVO$_4$ photoanodes. **d** Chopped $J$–$V$ curves of CoOOH/BP/BiVO$_4$ and MnO$_x$/BP/BiVO$_4$ photoanodes in KPi electrolyte (pH = 7.1) under AM 1.5 illumination

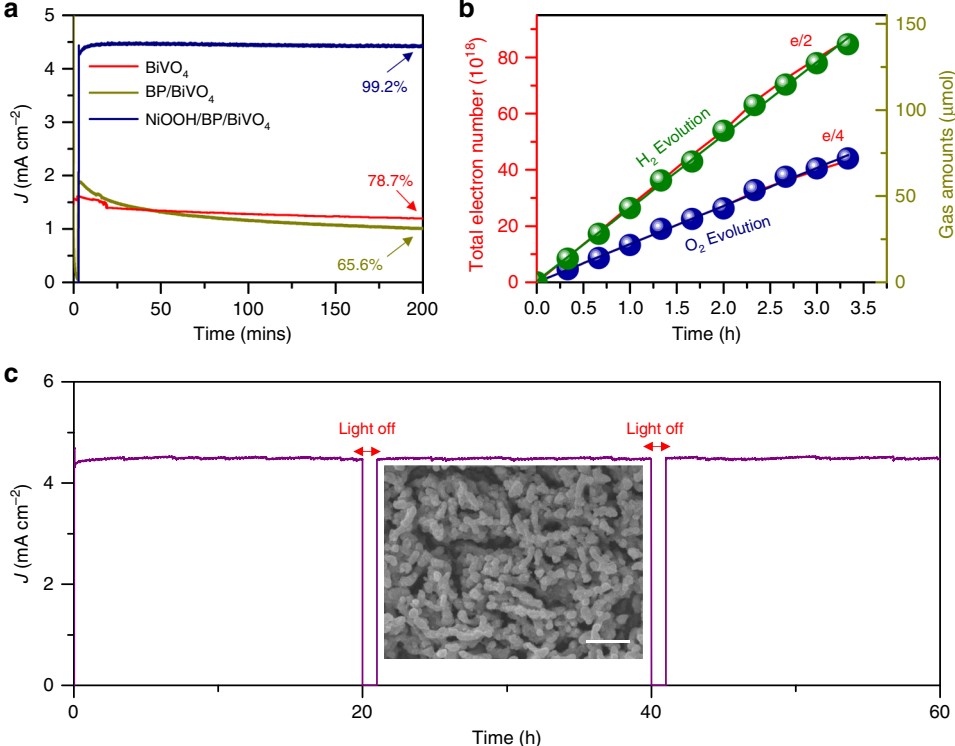

**Fig. 5 a** Photocurrent density stability measured at 1.23 V vs RHE in KPi electrolyte (pH = 7.1) under AM 1.5 illumination. **b** Plots of the theoretical charge number obtained from the $J$–$t$ curves collected at 1.23 V vs. RHE and the actual quantities of H$_2$ and O$_2$ evolution in KPi electrolyte (pH = 7.1) under AM 1.5 illumination. **c** Long-term stability of the NiOOH/BP/BiVO$_4$ photoanode at 1.23 V vs RHE in KPi electrolyte (pH = 7.1) under AM 1.5 illumination, insert is SEM image after long-term testing, scan bar: 1 μm

NiOOH/BP/BiVO$_4$ photoanode is stable with a slight fluctuation between 4.31 and 4.56 mA/cm$^2$. The morphology after testing is well-maintained (Insert in Fig. 5c). The result is a sharp contrast to the CoOOH/BP/BiVO$_4$ photoanode which shows a rapid decrease in photocurrent density after continuous testing of 2 h (Supplementary Fig. 27). The XPS results of NiOOH/BP/BiVO$_4$ photoanodes before and after long-term testing are shown in Supplementary Fig. 28, which indicates that the BP in the NiOOH/BP/BiVO$_4$ photoanode is oxidized to a lesser extent. However, the BP in the CoOOH/BP/BiVO$_4$ is almost completely oxidized (Supplementary Fig. 29). It is a fact that the BP is able to be slowly oxidized during PEC testing, whereas the electrodeposited NiOOH is believed to have conformal coverage which impedes the oxidization of BP.

## Discussion

In this study, we have demonstrated that a layer of BP nanosheets can serve as an excellent hole extraction layer in a BiVO$_4$/OEC photoanode for solar water splitting. The BP nanosheets, which were exfoliated from layered bulk BP, had the unique merit of $p$-type conductivity, hence enabling the formation of a $p/n$ heterojunction with BiVO$_4$, which facilitated hole transfer from BiVO$_4$ to the OEC surface. As a result, the NiOOH/BP/BiVO$_4$ photoanode exhibited a photocurrent density of 4.48 mA·cm$^{-2}$ at 1.23 V vs RHE under AM 1.5 illumination, which was 4.2 times higher than that of pure BiVO$_4$ and 1.5 times higher than that of NiOOH/BiVO$_4$. The BP layer was found to store separated holes and then transfer them to the OEC surface, and this impressive function was universal for other OECs, such as CoOOH and MnO$_x$. Moreover, the burying of the BP nanosheets by the OEC layer alleviated self-oxidation, thereby prolonging the stability of photoelectrochemical water splitting by BiVO$_4$. Our work shows the potential for application of BP in solar energy conversion devices, nevertheless, a uniform coating of BP on photoanodes with strongly coupled interface is still desired for further optimization.

## Methods

**Materials**. All chemical reagents were purchased from Aldrich without further purification. FTO was purchased from TEC-8, Pilkilton with a resistance of 14 Ω. BP crystal was purchased from Mukenano Co. LTD.

**Synthesis of BP/BiVO$_4$ Photoanodes**. BiVO$_4$ electrodes were prepared based on Lee and Chio's method[13]. Bulk BP crystals were exfoliated by ultrasonication in a mixture of γ-butyrolactone (GBL) and IPA. Briefly, 20 mg of BP crystals was dispersed into 20 mL of the mixture and sonicated for 10 h at 300 W. The resultant dispersion was centrifuged at 2000 rpm for 60 min The exfoliated BP sheets were dispersed again in IPA at a concentration of 0.02 mg/mL and stored under flowing N$_2$. The as-prepared BiVO$_4$ photoanode with a size of 1 × 2 cm$^2$ was placed against the wall of a 50 mL centrifuge tube with the sample side facing up. Then, 50 mL BP/IPA dispersion was added to the centrifuge tube and centrifuged by 1000 rpm for 1 min For self-absorption or deposition of BP on BiVO$_4$ photoanode, the as-prepared BiVO$_4$ photoanode with a size of 1 × 2 cm$^2$ was immersed in 10 mL of the BP sheet dispersion for 2 h in a glovebox. All BP/BiVO$_4$ photoanode was further dried at 50 °C in a vacuum oven.

**Synthesis of OEC/BP/BiVO$_4$ photoanodes**. The NiOOH and FeOOH layers were photoelectrodeposited on photoanodes by Lee and Chio's method[13]. Briefly, the NiOOH was photoelectrondeposited on BP/BiVO$_4$ and BiVO$_4$ photoanodes in a 0.1 M NiSO$_4$ solution with pH adjusted to 6.8 by carefully adding NaOH at 0.11 V vs. Ag/AgCl (total charge 22 mC cm$^{-2}$) under AM 1.5 illumination. The FeOOH was photoelectrodeposited on BiVO$_4$ photoanode in a 0.1 M FeSO$_4$ solution with gently stirring at 0.25 V vs. Ag/AgCl. The CoOOH layer was deposited on BP/BiVO$_4$ and BiVO$_4$ photoanodes by a spin coating method. The CoOOH ink was synthesis based on Huang's report[49]. Briefly, 15 mg CoCl$_2$·6H$_2$O was dissolved into 40 mL ethylene glycol, and the pH value of the solution was adjusted to 9.5 by slowly dropping 25% NH$_3$·H$_2$O. The mixture was then transferred to a Teflon-lined stainless steel autoclave with a total 60 mL capacity and maintained at 130 °C for 24 h. The obtained α-Co(OH)$_2$ nanosheets were dispersed in water/ethanol mixture with a 1:1 volume ratio after being washed with deionized water and ethanol several times. The pH value of greenish Co(OH)$_2$ suspension was adjusted to 12 by

adding 0.5 M NaOH solution. Then, a 5.2 wt% NaClO solution was slowly dropped into the suspension under vigorous stirring until the color changed to brown-black. The resulted CoOOH were obtained by ultrasonication assisted exfoliation for 12 h and was dispersed in alcohol to form a homogenous ink. CoOOH ink was then spin-coated onto a BiVO$_4$ electrode at 2000 r.p.m. for 1 min and dried at 50 °C vacuum oven. The MnO$_x$ layer was photo-deposited on BP/BiVO$_4$ and BiVO$_4$ photoanodes based on Li's report[50]. 5 mL 0.01 M MnSO$_4$ solution and 5 mL 0.02 M NaIO$_3$ solution were mixed in petri dish. BP/BiVO$_4$ and BiVO$_4$ photoanodes were placed in the above solution with the material side facing up under AM 1.5 illumination for 5 min Before PEC testing, the samples were under irradiation for 1–10 min, which can make open-circuit voltage achieve the best effect[14].

**Material Characterization**. SEM images of the products were recorded on a field-emission scanning electron microscope (JSM-7000F, Japan). The XRD patterns were obtained with a D500/5000 diffractometer operated in Bragg–Brentano geometry and equipped with a Cu-Kα radiation source. The HR-TEM observations were performed on a JEOL JEM-AFM 200F (Japan) electron microscope with (Cs-corrected/energy-dispersive X-ray spectroscopy (EDS)/EELS). The VB-XPS and XPS measurements were performed on an auger electron spectroscopy (AES) XPS instrument (ESCA2000 from VG Microtech in England) equipped with an aluminum anode (Al Kα, λ = 1486.6 eV). The UV–Vis DRS spectra were recorded using a UV–Vis spectrophotometer (Shimadzu UV-2550). Raman spectra were measured by using LabRam Aramis equipment (Horriba Jovin Yvon Inc., US). The AFM was measured by using Bruker Multimodel-8 equipment.

**In-situ time-resolved transient absorption spectroscopy**. The Laser flash photoelectrochemical water splitting measurements with transmission detection were performed with the third harmonic of the Nd;YAG laser (10Hz, NT342A, EKSPLA, 355 nm (3.5 mJ/pulse)) as excitation source and the Xe lamp (continuous wave, 300W, Newport) as the probe light source, in a three electrode system with working electrode (3 × 3 cm$^2$), Pt counter electrode and Ag@AgCl reference electrode, and N$_2$ saturated 0.5 M KPi buffer electrolyte. The transmitted probe light was focused on a monochromator (Princeton Instruments, Acton SpectraPro SP-2300). The output of the monochromator was monitored using a photomultiplier Tube (PDS-1, Dongwoo Optron). The transient signals were passed through an amplifier (SR445A, Stanford Research Systems) and then recorded by a digital oscilloscope (350MHz, MDO4034C, Tektronix). Photoanodes were placed in a sealed reactor with Argon purged phosphate buffer electrolyte. The applied bias was controlled with a PGSTAT204 potentiostat (Metro Autolab).

**Computational method**. Density functional theory calculations are performed using the plane-wave basis sets in VASP code[51]. The ion–electron interaction is treated by the projected-augmented wave (PAW) approximation[52]. The exchange-correlation functional is expressed with generalized gradient approximation Perdew−Burke−Ernzerhof (PBE-GGA)[52]. The energy cutoff for the plane-wave basis is set to 400 eV and the convergence threshold is set as 10$^{-4}$ eV in energy and 0.01 eVÅ$^{-1}$ in force. DFT+D2 method is adopted to describe the Grimme vdw correction during structure simulation and electronic calculation[53]. The Brillouin zone is set to 6 × 3 × 6 for bulk structure geometry optimization, 4 × 2 × 1 for BiVO$_4$/BP heterojunctions and 12 × 6 × 1 for electronic properties. The isosurface value for VBM and CBM is set to 0.002 e/Bohr$^3$.

**PEC Measurements**. The PEC performance was measured using a potentiostat (CH Instruments, CHI 660) in a three-electrode optical O-ring cell (0.37 cm$^2$) with a Pt foil counter electrode and a saturated Ag/AgCl reference electrode (in 3 M KCl) under AM 1.5G simulated solar light illumination (100 mW cm$^{-2}$) from a 150 W xenon lamp solar simulator (PEC-L01, PECCELL, Yokohama, Japan), all electrodes were illuminated from rear side. For comparison purpose, the front-illuminated performances were measured in Quartz reactor with an active area of 0.8 cm$^2$. In addition, Before the measurements, the solar simulator intensity was calibrated with a reference silicon solar cell (VLSI standards, Oriel P/N 91150 V). KH$_2$PO$_4$ and K$_2$HPO$_4$ buffer solution (pH = 7.1) with 0.5 M concertation was used as the electrolyte. The conversion between the potentials vs. Ag/AgCl and vs. RHE was performed using the following equations:

$$E(\text{vs RHE}) = E(\text{vs Ag/AgCl}) + E_{\text{Ag/AgCl}} + 0.0591 \times \text{pH} \qquad (2)$$

$$E_{\text{Ag/AgCl}} = 0.1976 \text{ vs. RHE.} \qquad (3)$$

Prior to the PEC measurements, the electrolyte was purged with N$_2$ to remove dissolved oxygen. In a typical J–V measurement, linear sweep voltammetry was conducted at a scan rate of 20 mV s$^{-1}$. The potentiostatic mode was used to measure the electrochemical impedance spectra (EIS) with an AC voltage amplitude of 5 mV and a frequency range of 0.01–100 kHz under AM 1.5G illumination. When doing the record, a silver paste was painted on the top to increase the conductivity and an aperture was used to determine the contact area between the samples and the electrolyte.

The gas evolution was carried out in a quartz reactor, which was sealed with rubber plugs and Parafilm. The electrode (1.5 cm$^2$) was immersed in the electrolyte in a three-electrode configuration with a 1.23 V vs RHE. Prior to the reaction and

the sealing process, the electrolyte was purged with $N_2$ gas. 1 mL of gas was analyzed by gas chromatography (Agilent Technologies 7890A GC system, USA) using a 5 Å molecular sieve column and Ar as the carrier gas. The experimental error for the evolution of $H_2$ and $O_2$ was considered to be ≈3%.

The theoretical electron number as a function of the $J$–$t$ curve was calculated on the basis of an area of 1.5 cm$^2$.

$$\text{Theoretical electron number} = \int_{t=0\,\text{min}}^{t=200\,\text{min}} \text{current density} \times 1.5 \times 6.24146 \times 10^{18} \tag{4}$$

The photocurrent-to-$H_2$ conversion efficiency and photocurrent-to-$O_2$ conversion efficiency were determined on the basis of their linear slopes (i.e., $\frac{\text{theoretical electron number}}{2}$ for the photocurrent-to-$H_2$ conversion efficiency and $\frac{\text{theoretical electron number}}{4}$ for the photocurrent-to-$O_2$ conversion efficiency).

**Calculation of the theoretical photocurrent in BiVO$_4$ photoanodes**. The single photon energy is calculated from Eq. (5)

$$E(\lambda) = h \times C/\lambda, \tag{5}$$

where E($\lambda$) is the photon energy (J), h is Planck's constant ($6.626 \times 10^{-34}$ Js), C is the speed of light ($3 \times 10^8$ m s$^{-1}$) and $\lambda$ is the photon wavelength (nm).

The solar photon flux is then calculated according to Eq. (6)

$$\text{Flux}(\lambda) = (\lambda)/E(\lambda), \tag{6}$$

where Flux($\lambda$) is the solar photon flux (m$^{-2}$ s$^{-1}$ nm$^{-1}$), and $P(\lambda)$ is the solar power flux (W m$^{-2}$ nm$^{-1}$).1 The theoretical maximum photocurrent density under solar illumination (AM1.5), $J_{max}$ (A m$^{-2}$), is then calculated by integrating the solar photon flux between 300 to 515 nm, shown in Eq. (7):

$$j_{max} = e \times \int_{300\,\text{nm}}^{515\,\text{nm}} \text{Flux}(\lambda)d\lambda, \tag{7}$$

where $e$ is the elementary charge ($1.602 \times 10^{-19}$ C). The theoretical photocurrent of such BiVO$_4$ photoanodes is accordingly calculated to be 6.87 mA cm$^{-2}$ based our solar spectra.

## Data availability
The data supporting the findings of this study are available within the article and its supplementary information files and from the corresponding author upon reasonable request.

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

## Acknowledgements

This work was supported by NSFC (51802157, 5151101197, 61725402), the Natural Science Foundation of Jiangsu Province of China (BK20180493), NRF Korea (NRF-2019R1A2C3010479, 2015M1A2A2074663, 2016M3D3A1A01913254 (C1 Gas Refinery Program)), the Fundamental Research Funds for the Central Universities (Nos. 30917011202, 30915012205, 30916015106), and PAPD of Jiangsu Higher Education Institutions. K.Z. acknowledges the support by "the Fundamental Research Funds for the Central Universities", No.30918011106. K.Z. and B.J.J. contributed equally to this work.

## Author contributions

K.Z. and J.H.P. conceived and designed the experiments. K.Z., B.J.J., and Y.J.C. carried out materials synthesis and electrochemical characterization. X.F.S. participated in part of the synthesis. X.J.S. participated in part of the materials synthesis. C.P. and W. Kim carried out TA measurement and analysis. S.L.Z. carried out theoretical simulation. K.Z. H.B.Z. and J.H.P. co-wrote the paper. All authors discussed the results and commented on the manuscript.
