## [Peer Review File · Nature Communications]

Reviewers' comments:

Reviewer #1 (Remarks to the Author):

In their manuscript, Zhang et al. probe the photoelectrochemical (PEC) performance of BiVO₄ photoanodes that use a black phosphorene (BP) layer as an interfacial layer between BiVO₄ and typical oxygen evolution catalysts (OECs). The use of BP as an interfacial layer is new, and shows interesting results (formation of a p-n junction) that could have interest in the PEC community. The overall performance is not very impressive, and makes only a small advance over other similar systems (that of Choi, Science), however, the approach is interesting (p-n junctions at the surface), and the material is new (BP), and so this may be suitable for consideration after significant revisions, detailed below.

I have several questions about coverage. The authors mention they optimize the system with 4 layers, but is this really 4 layers uniformly deposited across the 3D film, or will there still be some BiVO₄/electrolyte or BiVO₄/OEC interfaces that promote different charge separation/catalytic performance? Can the authors comment on the coverage of BP on the BiVO₄, not only laterally on the surface, but vertically within the 3D porous structure?

Where the samples illuminated from the front or back? Can the authors show the PEC response for both configurations to further improve the understanding of electron/hole transport limitations?

Can the authors comment on the morphology/composition of the composite materials after PEC testing? The mention of stability for ~100 minutes is nothing interesting, and separately, this should be extended for at least 10 hours, closer to 1 day to be able to even begin discussing stability. However, how do the films look, and what is the composition of the surface and interfaces after PEC testing?

In SI Figure 9, the authors say that the valence band is composed of O p and Bi p orbitals (should be orbitals), and that the conduction band is mainly V d orbitals. However, the figure shows that the VB is mostly O-p, and that the CB is V-d and O-p and Bi-p with contributions from O-s. Therefore the text seems not in line with the actual figure they produce. Can the authors fix or comment on this (and put in terms of other works done on DOS for monoclinic BiVO₄?

In SI Figure 11, the BP/BiVO₄ seems to be a worse dark electrocatalyst compared to pure BiVO₄. Furthermore, comparing Figure S11 and S12, the BP/BiVO₄ has very different characteristics, i.e. there is an onset potential > 2 V vs. RHE for figure S11, while there is an onset < 2 V vs. RHE for Figure S12. To me this points to inhomogeneities in the system (as to be expected when putting 2D materials on a 3D porous semiconductor substrate), and then calls into question the data presented in the main text, i.e. did the authors pick and choose particular results that showed results in line with trends they hoped to observe, or do the main text figures represent typical/averaged results of the samples with BP?

Figure S15 shows that the BP layer has absorption from 300 to 700 nm, but in the main text Figure 3(e), it seems that this is not enhancing absorption for the BiVO₄/BP composite in the same range. Can the authors comment on this discrepancy?

In Figure S21, the transients for the NiOOH/BP are more pronounced than the NiOOH. Can the authors discuss what this means for interfacial charge transfer mechanisms?

If the authors can address the previous questions, then the manuscript should be heavily revised, and re-evaluated before further consideration.

Reviewer #2 (Remarks to the Author):

Dear editor

The article titled "Black Phosphorene: A Hole Extraction Layer Boosting Solar Water Splitting beyond Oxygen Evolution Catalyst" reported that the built-in p/n electric field between BiVO₄ and BP strongly drives holes from BiVO₄ to the NiOOH surface for efficient water oxidation. BP is treated as a holes extraction layer to store separated holes. BP has achieved great triumph in the optic, electronic, catalytic, and energy conversion fields over the past decade. Although several works have demonstrated 2D BP coupled with metal oxide semiconductor, eg. BiVO₄ (J. Mater. Chem. A, 2018, 6, 19167–19175) (Angew. Chem. Int. Ed. 2018, 57, 2160–2164), TiO₂ (ChemElectroChem 2017, 4, 2373 – 2377), it is first time to proposing the conception of extraction layer for BP. The author compared it with OER like FeOOH, showing its excellent holes storage and transport properties via Transient absorption spectrum (TA) and electrochemical impedance spectroscopy (EIS). The texts and figures were well organized and logical. The conclusions were also supported by the experimental results. So I'd recommend to accept this manuscript after major revision.

1. The number of the figure in the supplementary material cannot correspond to the text one by one. Please check carefully.
2. In Figure S2, tell the reason for the obvious difference between the two SEM images? Write below the Figure S2
3. The author did not explicitly write the reasons for preparing BP using centrifugal coating method; add the PEC performance of BP coated BiVO₄ via Immersion deposition method
4. It can be seen from Fig. 1 that BP is not uniformly covered on BiVO₄, which has little effect on the system. Is there another way to evenly coat?
5. Is the horizontal axis of Figure S9 correct?
6. In Figure S 11, try to test to 5mA to see the slope or look at the Tafel curve.
7. The author should introduce the circuit components that fit the circuit diagram. Since BP has energy storage, why does the EIS spectrum of BP/BiVO₄ not form an impedance spectrum of another time constant (ie, an additional surface BP capacitor layer)?
8. In line 219, the article should list the EIS fit of BP and FeOOH.
9. At line 237, the authors said that a high photocurrent drop indicates a strong capacitive capacity that facilitates charge extraction. But it can be also said that the high photocurrent drop indicates that more holes accumulation, which is not conducive to charge transfer (Energy Environ. Sci., 2018, 11, 2972–2984). How to correctly distinguish between the two interpretations.
10. Please list the preparation methods of CoOOH and MnOx. All the used samples in the paper should be listed their preparation methods.
11. Please use a more distinguishing color in Figure 4d.
12. minor mistakes:
 - 1) Line 156, 41.97 change to 141.97.
 - 2) Should R3 of line 219 be changed to R2?

Reviewer #3 (Remarks to the Author):

This is a nice paper with a full suite of characterisation of the materials used. The results are impressive although the advantages over FeOOH don't seem to be so big?
Was a FeOOH/BP/BiVO₄ or even a NiOOH/BP/FeOOH/BiVO₄ or other combinations investigated?
I am very surprised by the stability of the BP systems—water and oxygen are the two species known to decompose phosphorene quickly. Longer run times would make the case more compelling.
The phosphorene sample used is poor quality. This is obvious from the cloudy dispersion obtained the uv-vis presented which is simply the scattering of the small particles produced in the long sonication

times.

I would recommend publication if a stronger case of the advantages of the BP and longer stabilites can be demonstrated.

Response to the Comments on Nature Communications

Manuscript ID: NCOMMS-18-36462

Dear Reviewers:

We appreciate the referee's comments and suggestions. We revised the manuscript thoroughly according to those comments. The added items are highlighted in red in the main text and the supplementary information. More detailed our response and changes were made and listed below:

Reviewer(s)' Comments to Author:

Reviewer #1: In their manuscript, Zhang et al. probe the photoelectrochemical (PEC) performance of BiVO₄ photoanodes that use a black phosphorene (BP) layer as an interfacial layer between BiVO₄ and typical oxygen evolution catalysts (OECs). The use of BP as an interfacial layer is new and shows interesting results (formation of a p-n junction) that could have an interest in the PEC community. The overall performance is not very impressive and makes only a small advance over other similar systems (that of Choi, Science), however, the approach is interesting (p-n junctions at the surface), and the material is new (BP), and so this may be suitable for consideration after significant revisions, detailed below.

Our Response: We thank very much for your positive evaluation of our work. The questions and suggestions raised by you are extremely important and helpful, which make us deep thought, thereby improving the quality of our work in the revision. As will be shown below, we have done a series of investigations on electrochemical properties, morphology, chemical component et.al. The main contribution of our work is further strengthened, and we believe the quality of the paper is significantly improved.

1#1: I have several questions about coverage. The authors mention they optimize the system with 4 layers, but is this really 4 layers uniformly deposited across the 3D film, or will there still be some BiVO₄/electrolyte or BiVO₄/OEC interfaces that promote different charge separation/catalytic performance? Can the authors comment on the coverage of BP on the BiVO₄, not only laterally on the surface, but vertically within the 3D porous structure?

Our Response: Thanks for your kind comments. As we can see in Figure 1, we could find that some BP could be coated on BiVO₄ grains. However, it is not easy to confirm whether all 3D porous BiVO₄ surface could be covered by 2D BP layers or not. According to the morphology characterizations, it must be admitted that the BP coated on BiVO₄ grain still remains BiVO₄/electrolyte or BiVO₄/OEC interfaces although we attempted the centrifugal coating approach to obtain more intimate interface between BP and BiVO₄ (please check our data; centrifugal coating vs. immersion deposition). By the TEM observation and electrochemical analysis, we can assume reasonably that most part of

BiVO_4 grains are coated by BP layers. In addition, we also have a few reservations on whether BP with 4 layers is best for hole extraction and storage behaviors since we could not control the layer number of BP herein. Nevertheless, we focused on the unique hole extraction behaviors of BP in this work. Regarding that the incorporation of BP and BiVO_4 for constructing p-n junction could be a potentially interesting direction, in further work, we are devoting to structure and performance optimizations, which are based on the unique hole extraction and storage behaviors of BP reported in here.

We understand your doubt that the poor coverage of BP on BiVO_4 surface or non-uniform structure (laterally on the surface or vertically within the 3D porous structure) may significantly detract from the work quality. To remove unwanted morphology complexity which may detract our work quality, we newly attempted another structure, which can maximize BP coverage on the BiVO_4 surface. More dense BiVO_4 film (Figure R1a) was synthesized by literature metal-organic decomposition method (J. Am. Chem. Soc. 2011, 133, 18370). Comparison of pure BiVO_4 (Figure R1b) and BP/ BiVO_4 (Figure R1c) from top-view SEM images determine conformal coverage of BP on the BiVO_4 surface. Both PEC performances are provided in Figure R2. Regardless of photocurrent density values, it can be clearly seen that the sufficient BP coverage can enhance PEC performance of BiVO_4 by ~2.1 times, the enhancement factor is remarkably higher than that presented in our manuscript. Furthermore, we took the absorbability into account because of the considerable proportion of BP, which is also possible to address the Q1#6 raised by you. As can be seen in Figure R3, the BP/ BiVO_4 electrode presents remarkably enhanced light harvesting efficiency in such thin film, which seems to be another responsible for the enhanced PEC performance.

In our revised manuscript, we mentioned the logic flow and pointed out the non-uniform coverage for its inadequacy in the discussion and conclusion parts. We totally agree and respect your criticism of the non-uniform interface.

Figure R1. (a) the cross-section SEM image of the almost single particle-thick BiVO_4 film, top view SEM image of single particle-thick BiVO_4 film (b) and BP covered film (c). (This figure is for reviewers only)

Figure R2. Chopped J - V curves in KPi electrolyte ($\text{pH}=7.1$) under AM 1.5 illumination. (This figure is for reviewers only)

Figure R3. LHE of BiVO_4 and BP/BiVO_4 photoanodes. (This figure is for reviewers only)

1#2: Where the samples illuminated from the front or back? Can the authors show the PEC response for both configurations to further improve the understanding of electron/hole transport limitations?

Our Response: Thanks for your wise comments. As described in our original manuscript, “The PEC performance was measured using a potentiostat (CH Instruments, CHI 660) in a three-electrode optical “O-ring” cell (0.37 cm^2) with a Pt foil counter electrode and a saturated Ag/AgCl reference electrode (in 3 M KCl) under AM 1.5G simulated solar light illumination (100 mW cm^{-2}) from a 150 W xenon lamp solar simulator (PEC-L01, PECCELL, Yokohama, Japan).”, the “O-ring” cell only allows back illumination, which was noted in our revised manuscript.

In addition, we measured the PEC performances of BiVO_4 and BP/BiVO_4 under front illumination. As shown in Figure R4a, the obtained photocurrent densities under front illumination in either KPi or $\text{Na}_2\text{SO}_3 + \text{KPi}$ electrolytes were lower than that under back illumination. Presumably, the nanoporous structure with insufficient transparency led to the photogenerated electrons which have a longer distance to reach FTO (Adv. Energy Mater. 2016, 6, 1501645). Correspondingly, the calculated η_{sep} for both BiVO_4 and BP/BiVO_4 are remarkably lower than the back illumination (Figure R4b), whereas the calculated η_{tran} for both BiVO_4 and BP/BiVO_4 under both side illumination are similar (Figure R4c).

Moreover, under back illumination, the η_{sep} of BP/BiVO_4 is slightly higher than that of BiVO_4 , which is similar to the results obtained under front illumination. However, the η_{tran} of BP/BiVO_4 displays different tendency from back illumination, where the η_{sep} of BP/BiVO_4 is also higher than that of BiVO_4 . The enhanced η_{tran} of BP/BiVO_4 photoanode from front illumination is unexpected, which might be ascribed to the effect of surface passivation by BP layers on the reduction of surface recombination that compensates poor water oxidation ability (J. Phys. Chem. C. 2015, 119, 7275-7281).

Figure R4: (a) J - V curves of BiVO_4 and BP/BiVO_4 measured in KPi electrolyte with and without hole scavenger ($0.5\text{M Na}_2\text{SO}_3$) under rear illumination. The corresponding charge separation efficiencies (b) and charger transfer efficiencies (c). (This Figure is included as Figure S21 in the revised manuscript).

1#3: Can the authors comment on the morphology/composition of the composite materials after PEC testing? The mention of stability for ~100 minutes is nothing interesting, and separately, this should be extended for at least 10 hours, closer to 1 day to be able to even begin discussing stability. However, how do the films look, and what is the composition of the surface and interfaces after PEC testing?

Our Response: Thanks for your nice question. Actually, the BiVO_4 composition is not completely stable during PEC measurements, as the BiVO_4 is able to be dissolved by the surface accumulated holes in aqueous solution (Nat. Energy 2016, 2, 16191; Nat. Energy 2018, 3, 53-60). In addition to the photo-corrosion of BiVO_4 , the easily oxidized BP on its surface would aggravate the instability. As a result, both BP/BiVO_4 and pure BiVO_4 presented poor stabilities, as demonstrated in Figure 5a in our original manuscript. Surface coated NiOOH is believed to act as a protection layer to suppress both photo-corrosion of BiVO_4 and oxidization of BP, which is obviously reflected by 200 min illumination. However, we totally agree with your criticism on that such a short time is not enough to discuss its stability. Therefore, in our revised manuscript, we performed the stability as long as possible (60 hours) for the demonstration of particle application, and investigated its morphology/composition changes during the long-term testing.

Figure R5: Long stability testing of NiOOH/BP/BiVO₄ in KPi electrolyte (pH=7.1) under AM 1.5 illumination (This Figure is included as Figure 5c in the revised manuscript).

Figure R5 is the long-term stability with 60 hours illumination at a bias of 1.23 V vs RHE, the steady Faradaic current for NiOOH/BP/BiVO₄ is stable with slight fluctuation, which can be considered as the contribution of NiOOH protection layer. Moreover, the morphology after the long-term testing is well-maintained (Figure R6).

Figure R6: SEM image of NiOOH/BP/BiVO₄ after long-term testing (This Figure is included as an inset of Figure 5c in the revised manuscript).

Furthermore, we performed the XPS investigations on the composition of the surface and interfaces after the long-term testing. As shown in Figure R7a, the O1s peak shows detectable changes in intensity and position, whereas a slight shift of the V 2p towards lower binding energy and reduced intensity can be observed after long-term testing. The phenomenon can be tracked to the photo-charging of BiVO₄ (Energy Environ. Sci. 2017, 10, 1517–1529; J. Mater. Chem. A 2016, 4, 2919–2926), by which the V in 5+ state was

somehow reduced to 4+ state. Figure R7b shows the P 2p peak before and after long-term testing. It is a fact that BP is able to be slowly oxidized during PEC testing, as evidenced by the raised P_xO_y peak. Nevertheless, the Ni 2p peaks were determined to be stable during PEC testing (Figure R7c). Overall, the BP/BiVO₄ protected by NiOOH layer can be relatively stable for at least 60 hours.

Figure R7: Comparisons of XPS results of NiOOH/BP/BiVO₄ before and after long-term testing. (a) V2p and O1s, (b) P2p and (c) Ni2p (This Figure is included as Figure S28 in the revised manuscript).

1#4: In SI Figure 9, the authors say that the valence band is composed of O p and Bi p orbits (should be orbitals), and that the conduction band is mainly V d orbits. However, the figure shows that the VB is mostly O-p, and that the CB is V-d and O-p and Bi-p with contributions from O-s. Therefore the text seems not in line with the actual figure they produce. Can the authors fix or comment on this (and put in terms of other works done on DOS for monoclinic BiVO₄?

Our Response: Thanks for your kind advice, according to which we have provided a new picture with overall orbitals (Figure R8) and rewritten this part as below: “The theoretical results demonstrate that the valence band is mainly contributed by O-p orbital, together with a small contribution from Bi-s orbital mixing. The conduction band is primarily of V-d, O-p, and Bip with contributions from O-s orbital.” Our results agree with the previous report on DOS for monoclinic BiVO₄ (Figure R9; Chem. Mater. 2014, 26, 5365–5373).

Figure R8: Partial density of states and the corresponding charge density of valence band maximum and conduction band minimum of bulk BiVO_4 (This Figure is included as Figure S10 in the revised manuscript).

Figure R9: Density of states (DOS) and partial DOS of monoclinic scheelite BiVO_4 projected onto atomic wave functions produced from references. (This figure is for reviewers only)

1#5: In SI Figure 11, the BP/BiVO₄ seems to be a worse dark electrocatalyst compared to pure BiVO₄. Furthermore, comparing Figure S11 and S12, the BP/BiVO₄ has very different characteristics, i.e. there is an onset potential > 2 V vs. RHE for figure S11, while there is an onset < 2 V vs. RHE for Figure S12. To me this points to inhomogeneities in the system (as to be expected when putting 2D materials on a 3D porous semiconductor substrate), and then calls into question the data presented in the main text, i.e. did the authors pick and choose particular results that showed results in line with trends they hoped to observe, or do the main text figures represent typical/averaged results of the samples with BP?

Our Response: Thanks for your nice question. We have double-checked our data that was provided in Figure S11 and S12. We found a mistake when we normalized their current densities. For all electrochemical and photoelectrochemical performances, the area measured was 0.37 cm² in the “O-ring” cell. However, when we normalized the CV curve of BP/BiVO₄ for Figure S12, the real current density divided 0.37 cm² by two times that arise from our operating mistake of origin software. To make the convincing comparison, we have corrected this mistake and grouped the results obtained by similar samples, including BP/BiVO₄ of Figure S11 and S12, NiOOH/BP/BiVO₄ of Figure S12 and S23. As shown in Figure R10a and b, similar trends can be confirmed. We are sorry for this mistake caused by our carelessness.

Figure R10: Reproduced dark current density of BP/BiVO₄ from Figure S11 and S12 (a) and NiOOH/BP/BiVO₄ from Figure S12 and S23 (b). (This figure is for reviewers only)

1#6: Figure S15 shows that the BP layer has absorption from 300 to 700 nm, but in the main text Figure 3(e), it seems that this is not enhancing absorption for the BiVO₄/BP composite in the same range. Can the authors comment on this discrepancy?

Our Response: Thanks for your nice question. We considered that this would be a very interesting theme. Actually, as shown in Figure R11, the BP layer remarkably increases the absorbability of BiVO₄ electrode in specific wavelength regions. As can be seen in Figure R11, the polycrystalline BiVO₄ film measured in our work has two clear shoulders, at about 308 and 453 nm respectively. The two shoulder bands are ascribed to the charge-transfer transition involving the V-O component and Bi and V centers (J. Phys. Chem. C, 2008, 112, 6099; Adv. Funct. Mater. 2006, 16, 2163). According to our theoretical simulations (Figure 2d and e) and experimental results (Figure 2a, b, and c), the interaction between BiVO₄ and BP is only taken place at O and Bi orbits, not V orbit. Therefore, we tentatively deduce that the second shoulder starting from 453 nm maintains the net charge-transfer transition of BiVO₄. Outside the absorption range of BiVO₄ (>515 nm), the enhanced absorbability of BP/BiVO₄ electrode can be purely ascribed to the presence of BP layers.

Figure R11: UV-vis absorbance of BiVO₄ and BP/BiVO₄ photoanodes. (This figure is same as Figure 3e)

Furthermore, we provided the reflectance (Figure R12a) and transmittance (Figure R12b) of BiVO₄ and BP/BiVO₄ that were used to calculate their absorbance. It can be clearly seen, in absorption region of BiVO₄ that the introduction of BP doesn't have any effect on

the light reflection of BiVO₄ photoanode, but significantly shifts the light transmission of the first shoulder of BiVO₄ photoanode.

Figure R12: UV-vis reflectance and transmittance of BiVO₄ and BP/BiVO₄ photoanodes. (This figure is for reviewers only)

1#7: In Figure S21, the transients for the NiOOH/BP are more pronounced than the NiOOH. Can the authors discuss what this means for interfacial charge transfer mechanisms?

Our Response: Thanks for your nice question. The transient photocurrent response is considered as holes accumulation on the surface of BiVO₄, which usually is detrimental for charge transfer kinetics (Energy Environ. Sci., 2011, 4, 958–964; Energy Environ. Sci., 2018, 11, 2972–2984). Therefore, in our paper, we are very careful to claim the superior hole extraction, mainly involving hole accumulation and storage, instead of the enhancement in charge transfer kinetics. In Figure S21, we emphasized on the hole extraction behaviors of BP from the band alignment of p-n junction at the applied bias smaller than 0.6 V vs RHE. The extracted holes reaching the surface participate the OER merely at high potentials when the injection barrier no longer impedes the charge transfer from the electrode to the electrolyte. More detail discussion regarding the hole extraction was answered in Q2#9.

Reviewer #2: The article titled “Black Phosphorene: A Hole Extraction Layer Boosting Solar Water Splitting beyond Oxygen Evolution Catalyst” reported that the built-in p/n electric field between BiVO₄ and BP strongly drives holes from BiVO₄ to the NiOOH surface for efficient water oxidation. BP is treated as holes extraction layer to store separated holes. BP has achieved great triumph in the optic, electronic, catalytic, and energy conversion fields over the past decade. Although several works have demonstrated 2D BP coupled with the metal oxide semiconductor, eg. BiVO₄ (J. Mater. Chem. A, 2018, 6, 19167–19175) (Angew. Chem. Int. Ed. 2018, 57, 2160–2164), TiO₂ (ChemElectroChem 2017, 4, 2373–2377), it is first time to proposing the conception of extraction layer for BP. The author compared it with OER like FeOOH, showing its excellent holes storage and transport properties via Transient absorption spectrum (TA) and electrochemical impedance spectroscopy (EIS). The texts and figures were well organized and logical. The conclusions were also supported by the experimental results. So I'd recommend accepting this manuscript after major revision.

Our Response: We thank very much for your positive evaluation of our work. The questions and suggestions raised by you are extremely important and helpful, which make us deep thought, thereby improving the quality of our work. As will be shown below, we have polished our manuscript towards higher quality through significant modifications based on your suggestions. The main contribution of our work is further strengthened, and we believe the quality of the paper is significantly improved.

2#1: The number of the figure in the supplementary material cannot correspond to the text one by one. Please check carefully.

Our Response: Thanks for your kind remaining. We have double-checked the number of Figures, ensuring that all Figures are well-corresponded to the discussions of the main text.

2#2: In Figure S2, tell the reason for the obvious difference between the two SEM images? Write below the Figure S2.

Our Response: Thanks for your kind remaining. We added the description regarding the morphology changes (presented in Figure S5 in supplement).

“The top-view SEM images indicate that the pure BiVO₄ photoanode has larger space from micropores than BP/BiVO₄. The changes in the porous electrode can be considered as the collapse of the structure to some extent that arises from high centripetal force through centrifuged deposition.”

2#3: The author did not explicitly write the reasons for preparing BP using centrifugal coating method; add the PEC performance of BP coated BiVO₄ via Immersion deposition method.

Our Response: Thanks for your kind remaining. First, we added a TEM image of BP/BiVO₄ prepared via immersion deposition method, which shows the poor connection between the BiVO₄ particle and BP sheets (Figure R13).

Figure R13: HAADF-STEM-EDX element mapping of BP/BiVO₄ obtained by immersion deposition method (This Figure is included as Figure S9 in the revised manuscript).

Second, we compared the PEC performances of the BP/BiVO₄ photoanodes prepared by a centrifugal coating method and immersion deposition method (Figure R14). As expected, the poor connection between the BP layer and BiVO₄ particle in the BP/BiVO₄ photoanodes prepared by immersion deposition method produces lower PEC performance than that prepared by a centrifugal coating method. Despite the meaningful results, we did not discuss this issue in the revised manuscript because it can disturb the logical flow of the content of the manuscript. Therefore, we would like to maintain the structure of the introduction section of the original manuscript as it is, and only provide the characterization of its interface.

Figure R14: Comparison of chopped J-V curves of BP/BiVO₄ photoanodes, respectively obtained by a centrifugal coating method and immersion deposition method. (This figure is for reviewers only)

2#4: It can be seen from Fig. 1 that BP is not uniformly covered on BiVO₄, which has little effect on the system. Is there another way to evenly coat?

Our Response: As answered in Q1#1, an even coating has higher possibility to further promote the positive effect on PEC performance. However, the chemical sensibility of BP and structural incompatibility between the 2D sheet and porous electrode set an insurmountable barrier to counterbalance the uniform coverage structure and impressive performance. We need further study to find new coating method for further improvement of the positive effects of BP. Thanks for your nice question and concern for such problem, it arouses our deep thought.

2#5: Is the horizontal axis of Figure S9 correct?

Our Response: There is a mistake of missing the x-axis label, and we have added it. The horizontal axis describes the number of states per interval of energy at each energy level available to be occupied in bulk BiVO₄. The horizontal axis represents the energy of electrons, where the Fermi level is shifted to 0 eV. The vertical axis represents the number of states. The DOS of BiVO₄ is calculated based on PBE method. In addition, the valence band exceeds the Fermi level (0 eV) due to the smearing of DOS. We recalculated the DOS of bulk BiVO₄ by using smaller smearing, as shown in Figure S10 at present.

2#6: In Figure S11, try to test to 5mA to see the slope or look at the Tafel curve.

Our Response: Thanks for your kind advice. We have measured the dark current density of BiVO₄ and BP/BiVO₄ photoanodes with wide applied bias and calculated the Tafel slope for understanding OER reaction. As shown in Figure R15, the estimated Tafel slope for BiVO₄ photoanode (268 mV dec⁻¹) is smaller than that for the BP/BiVO₄ photoanode (301 mV dec⁻¹), implying that the BP somewhat hinders the reaction kinetics of BiVO₄.

Figure R15: Dark current density (a) and the corresponding Tafel plots (b) of BiVO₄ and BP/BiVO₄ with a scanning rate of 2 mV/s (This Figure is included as Figure S12 in the revised manuscript).

2#7: The author should introduce the circuit components that fit the circuit diagram. Since BP has energy storage, why does the EIS spectrum of BP/BiVO₄ not form an impedance spectrum of another time constant (ie, an additional surface BP capacitor layer)?

Our Response: Thanks for your nice question. For the interpretation of PEC measurements of illuminated electrodes, we adopt a classical view that is depicted in Figure R16. Such equivalent circuit allows us to analyze the hole behaviors at the electrode/electrolyte interface, where the C_{trap} represents the component for hole accumulation. As reported previously (J. Am. Chem. Soc. 2012, 134, 4294–4302; J. Phys. Chem. Lett. 2012, 3, 2517–2522), two types of kinetic processes can be involved into this system: 1) trap-assisted recombination of electrons that determined by electron detrapping; 2) charge

transfer of holes from the surface traps. If the electron detrapping is far slower than charge transfer of holes from the traps, the EIS spectrum will display two semicircles, rather, the EIS spectrum will display one semicircle. Therefore, for either BiVO_4 or BP/BiVO_4 , the charge transfer of holes from surface traps, eg: surface defects or BP, is very slow, as determined by their η_{tran} . This is the reason that the EIS spectra of BiVO_4 and BP/BiVO_4 presented one semicircle.

Figure R16: Equivalent circuit corresponding to the physical model. (This figure is for reviewers only)

2#8: In line 219, the article should list the EIS fit of BP and FeOOH .

Our Response: Thanks for your kind reminding. Unfortunately, we could not totally understand your meaning. But we found an error of missing FeOOH word in line 219, we have corrected it.

2#9: At line 237, the authors said that a high photocurrent drop indicates a strong capacitive capacity that facilitates charge extraction. But it can be also said that the high photocurrent drop indicates that more holes accumulation, which is not conducive to charge transfer (Energy Environ. Sci., 2018, 11, 2972—2984). How to correctly distinguish between the two interpretations.

Our Response: Thanks for your wise advice and nice question. We agree with you that the transient photocurrent response is not conducive to charge transfer. In detail, the cathodic current transient is an indication of holes accumulation, while the anodic current transient represents the back reaction of electrons from the conduction band with the accumulated holes (J. Electrochem. Soc., 1981, 128, 2128–2133). To obtain steady Faradaic currents, the prevalent strategy is to eliminate those surface trapping state for holes accumulation by

introducing a passivation layer, referring to *Energy Environ. Sci.*, 2018, 11, 2972–2984. Another possibility is to improve the holes accumulation and delay the back reaction, which is called “hole extraction”. For the latter, the holes should be stored for a while (*Angew. Chem. Int. Ed.* 2014, 53, 7295–7299). Rather, if the materials for hole storage have good OEC activity, it will be an ideal model to significantly improve PEC performance. Unfortunately, the BP is not that one, thereby requiring additional OEC layer in our case. As such, the NiOOH/BP/BiVO₄ displayed the purely Faradaic photocurrent that is higher than those reference samples, although the large photocurrent drop still can be observed. In our original manuscript, finally, we ascribed the result to “**The fate of the holes extracted is to reach the surface, then participates in the water oxidation reaction at high potentials, where the injection barrier no longer impedes the charge transfer from the electrode to the electrolyte. These results further demonstrate the strong capability of the BP nanosheet layer to perform hole extraction towards water oxidation.**”.

2#10: Please list the preparation methods of CoOOH and MnO_x. All the used samples in the paper should be listed their preparation methods.

Our Response: Thanks for your kind advice. The preparation methods of CoOOH and MnO_x were provided in our revised manuscript.

“The CoOOH ink was synthesized based on Huang’s report (*Angew. Chem., Int. Ed.* 2015, 54, 8722). **Briefly, 15 mg CoCl₂·6H₂O was dissolved into 40 mL ethylene glycol, and the pH value of the solution was adjusted to 9.5 by slowly dropping 25% NH₃·H₂O. The mixture was then transferred to a Teflon-lined stainless steel autoclave with a total 60 mL capacity and maintained at 130 °C for 24 h. The obtained α-Co(OH)₂ nanosheets were dispersed in water/ethanol mixture with a 1:1 volume ratio after being washed with deionized water and ethanol several times. The pH value of greenish Co(OH)₂ suspension was adjusted to 12 by adding 0.5 M NaOH solution. Then, a 5.2 wt% NaClO solution was slowly dropped into the suspension under vigorous stirring until the color changed to brown-black. The resulted CoOOH were obtained by ultrasonication assisted exfoliation for 12 h and was dispersed in alcohol to form a homogenous ink. CoOOH ink was then spin-coated onto a BiVO₄ electrode at 2000 rpm for 1 min and dried at 50 °C vacuum oven. The MnO_x layer was photo-deposited on BP/BiVO₄ and BiVO₄ photoanodes based on Li’s**

report (*Nat. Commun.* 2013, 4, 1432). 5 mL 0.01 M MnSO_4 solution and 5 mL 0.02 M NaIO_3 solution were mixed in petri dish. BP/ BiVO_4 and BiVO_4 photoanodes were placed in the above solution with the material side facing up under AM 1.5 illumination for 5 min.”

2#11: Please use a more distinguishing color in Figure 4d.

Our Response: Thanks for your kind advice. Because the NiOOH cases were already presented in Figure 3d, in our revised manuscript, we removed the repeated result from Figure 4d. As shown in Figure R17, the presented curves are clear.

Figure R17: Reproduced J - V curves of CoOOH/BP/BiVO_4 and $\text{MnO}_x/\text{BP/BiVO}_4$ photoanodes in KPi electrolyte (pH=7.1) under AM 1.5 illumination (This Figure is included as Figure 4d in the revised manuscript).

2#12: minor mistakes: 1) Line 156, 41.97 change to 141.97. 2) Should R3 of line 219 be changed to R2?

Our Response: Thanks for your careful reading. We have corrected those errors.

Reviewer #3: This is a nice paper with a full suite of characterization of the materials used. The results are impressive although the advantages over FeOOH don't seem to be so big. I would recommend publication if a stronger case of the advantages of the BP and longer stability can be demonstrated.

Our Response: We thank very much for your positive evaluation of our work. The questions and suggestions raised by you are extremely important and helpful, which inspired us to further optimize or extend our finding for broad interests. According to your suggestions, we have strengthened the advantages of BP for the long-term PEC stability through a careful investigation, and we believe the quality of the paper is significantly improved.

3#1: Was a FeOOH/BP/BiVO ₄ or even a NiOOH/BP/FeOOH/BiVO ₄ or other combinations investigated?
--

Our Response: Thanks for your wise advice. Frankly, we didn't consider a NiOOH/BP/FeOOH/BiVO₄ photoanode, because the heterointerface of BP and BiVO₄ is central to our work. Also, we were unwilling to design a FeOOH/BP/BiVO₄ because NiOOH is more active for water oxidation and they are subjected to the similar electro-deposition procedure. Actually, we prepared a NiOOH/FeOOH/BP/BiVO₄ photoanode, which was expected to have better performance. However, the NiOOH/FeOOH/BP/BiVO₄ photoanode displays deteriorated photocurrent density at high voltage region, where a dramatic transient photocurrent response can be observed (Figure R18). The unexpected phenomenon is very similar to photoanode with multiple-charge transfer interface (Sustainable Energy Fuels, 2018, 2, 1979-1985; J. Mater. Chem. A, 2017, 5, 9952–9959; Appl. Catal. B: Environ. 2018, 237, 763–771), which should not be simply attributed to a single cause but rather a complex effect on charge transport, probably, Fermi-energy pinning. Of various factors affecting PEC performances, the composition complexity will dilute the role of BP. Therefore, we would like to maintain the logic flow and research focusing on this time.

Figure R18: J - V curves of NiOOH/BP/BiVO₄ and NiOOH/FeOOH/BP/BiVO₄ photoanodes in KPi electrolyte (pH=7.1) under AM 1.5 illumination. (This figure is for reviewers only)

3#2: I am very surprised by the stability of the BP systems—water and oxygen are the two species known to decompose phosphorene quickly. Longer run times would make the case more compelling.

Our Response: As is a similar concern to Q1#3, we tested the stability of NiOOH/BP/BiVO₄ for 60 hours.

Figure R5: Long stability testing of NiOOH/BP/BiVO₄ in KPi electrolyte (pH=7.1) under AM 1.5 illumination (This Figure is included as Figure 5c in the revised manuscript).

Figure R7: Comparisons of XPS results of NiOOH/BP/BiVO₄ before and after long-term testing. (a) V2p and O1s, (b) P2p and (c) Ni2p (This Figure is included as Figure S28 in the revised manuscript).

Obvious oxidization of BP can be detected, whereas its PEC performance has very small fluctuation during at least 60 hours. We believed that the reason for stability has complicated since there are the concurrent changes in V valence state and BP oxidation. Nevertheless, it is a fact that the uniformly deposited NiOOH well-impeded both BiVO₄ corrosion and BP oxidation to a large extent. We also tested the stability of CoOOH/BP/BiVO₄, which was prepared by spin-coating. Because the spin-coating of 2D CoOOH sheets exposed more surface of the BP surface, the Faradaic current displays a gradual decay from the beginning, and a rapid fading after 2 hours (Figure R19a). P2p XPS result indicates that BP is almost completely oxidized (Figure R19b). The results illustrate that BP sheets can be on active service for PEC water splitting with a perfect protection layer.

Figure R19: (a) Photocurrent density stability of CoOOH/BP/BiVO₄ photoanode measured at 1.23 V vs RHE in KPi electrolyte (pH=7.1) under AM 1.5 illumination (This Figure is included as Figure S27 in the revised manuscript), (b) P2p XPS spectrum of

CoOOH/BP/BiVO₄ before and after long-term testing (This Figure is included as Figure S29 in the revised manuscript).

3#3: The phosphorene sample used is poor quality. This is obvious from the cloudy dispersion obtained the uv-vis presented which is simply the scattering of the small particles produced in the long sonication times.

Our Response: We agree with you that the exfoliation of BP sheets is based on the previous reporting without further modification (Adv. Funct. Mater. 2016, 26, 2016–2024). Actually, the layer number of BP exfoliated through various methods can be well controlled at present. Because of the layer-dependent band gap of BP, it is expected to see more works with respect to BP/semiconductor heterojunction in the future, partly, motivated by this work. Thanks for your positive comments on our manuscript.

Reviewers' comments:

Reviewer #1 (Remarks to the Author):

The authors are thanked for their detailed response to earlier reviewer comments. I still have some questions about mechanistic interpretation.

In the new results provided in the rebuttal, particularly figure R2, there seems to be a discrepancy. While the addition of BP does indeed increase the photocurrent density of BiVO₄ across the whole potential range, the onset potential does not change at all. This is strange because the mechanism for enhancement (improved catalysis by BP and/or charge separation via p-n junction at the surface) should lower the onset potential for OER. How do the authors explain an increased photocurrent density without a lowered onset potential (especially in terms of the mechanism proposed for how BP improves PEC performance)?

Also, when looking at R3, it seems that the BP absorbs across the whole spectrum that BiVO₄ does, so is this just a dual absorber effect? However, in figure R11, the BP absorption is outside the band gap. Can BP be used by itself as a photoelectrode? Did the authors try this? How do the authors compare figures R3 and R11?

Other than this, the authors significantly improved the manuscript and addressed many questions well from all 3 reviewers. If the authors can address these final issues, then it is scientifically more sound and would warrant publication.

Reviewer #2 (Remarks to the Author):

Dear editor and author

The three reviewers have put forward constructive comments, which is very beneficial to the technical improvement of this article. After careful revision by the author, all questions have been answered more rigorously, and the article has become more mature and conducive to publication. So I'd recommend this article to accept after addressing the below minor question:

please show me the value of C_{bulk} and C_{trap} (especially C_{trap}) to see whether charge storage is positively correlated with the fitted surface capacitance.

Reviewer #3 (Remarks to the Author):

My concerns have been covered and I am happy to recommend publication.

Response to the Comments on Nature Communications

Manuscript ID: NCOMMS-18-36462A

Dear Reviewers:

We would like to thank the reviewers for careful reading and helpful comments. We revised the manuscript thoroughly according to the comments. The added items are highlighted in red in the main manuscript and the supplementary information. Following changes were made and listed below:

Reviewer(s)' Comments to Author:

Reviewer #1: The authors are thanked for their detailed response to earlier reviewer comments. I still have some questions about mechanistic interpretation. Other than this, the authors significantly improved the manuscript and addressed many questions well from all 3 reviewers. If the authors can address these final issues, then it is scientifically more sound and would warrant publication.

Our Response: We thank for your highly positive comments and many suggestions to improve our paper.

1#1: In the new results provided in the rebuttal, particularly figure R2, there seems to be a discrepancy. While the addition of BP does indeed increase the photocurrent density of BiVO₄ across the whole potential range, the onset potential does not change at all. This is strange because the mechanism for enhancement (improved catalysis by BP and/or charge separation via p-n junction at the surface) should lower the onset potential for OER. How do the authors explain an increased photocurrent density without a lowered onset potential (especially in terms of the mechanism proposed for how BP improves PEC performance)?

Our Response: Thanks for your kind comments. We agree with you that the onset potential of photoelectrode is mainly dominated by two factors, surface state passivation or loading of oxygen evolution co-catalyst for a photoelectrode. In our case, the BP does not act the role of co-catalyst for oxygen evolution, which in turn lowers water oxidation

kinetics to some extent (Figure R21). Another with respect to charge separation, the BP, as either photoabsorber (related to Q1#2) or hole extraction layer, should be responsible for a cathodic shift of onset potential. We understand your doubt about it because the onset potential shift is undetectable in Figure R2 that we provided in the last response letter. Actually, visualizing is not real. We re-draw our Figure by the enlarged y-axis, as shown in Figure R22. It can be clearly seen that only transient photocurrent responses with no steady Faradaic currents were observed for BiVO₄ anode at potential ranging from 0.1 to ~0.5 V vs RHE (anodic photocurrent density is approximate to cathodic photocurrent density), ascribing to poor charge separation. By contrast, the anodic photocurrent density of BP/BiVO₄ photoanode is larger than the cathodic photocurrent density at such potential region, indicating a lowered onset potential. A similar phenomenon can be tracked in many photoanodes with Chopped light chronoamperometry measurements, where the anodic and cathodic photocurrents are earlier occurrence than its onset potential (Chem. Sci., 2017, 8, 3712; Phys. Chem. Chem. Phys., 2013, 15, 4589; Energy Environ. Sci., 2014, 7, 1402; J. Mater. Chem. A, 2015, 3, 20649; J. Electroanal. Chem. 2014, 716, 8; J. Phys. Chem. C 2012, 116, 14541).

Figure R21: Dark current density (a) and the corresponding Tafel plots (b) of BiVO₄ and BP/BiVO₄ with a scanning rate of 2 mV/s (This Figure is included as Figure S12 in the revised manuscript).

Figure R22: Reproduced J-V curves of BiVO₄ and BP/BiVO₄ photoanodes in KPi electrolyte (pH=7.1) under AM 1.5 illumination (This Figure is for reviewers only).

1#2: Also, when looking at R3, it seems that the BP absorbs across the whole spectrum that BiVO₄ does, so is this just a dual absorber effect? However, in figure R11, the BP absorption is outside the band gap. Can BP be used by itself as a photoelectrode? Did the authors try this? How do the authors compare figures R3 and R11?

Our Response: Thanks for your nice question. For Figure R3, we more like to believe that this is optical interference in a transparent film. Because the BiVO₄ electrode measured for Figure R3 is very thin (50~60 nm), its UV-vis absorbance exhibited characteristic glass absorption peaks, instead of BiVO₄ or BiVO₄/BP (Adv. Energy Mater. 2016, 6, 1501645). To elucidate the BP effect on absorption of BiVO₄, we prepared thicker BiVO₄ films for observing the absorption edge of BiVO₄ and controlled BP content for comparing the absorption enhancement. Notes that the amount of BP in BP-2/BiVO₄ photoanode is two times higher than that in BP-1/BiVO₄ photoanode. As shown in Figure R23, the BP absorbs across the whole spectrum that BiVO₄ does in the absorption spectrum of BP-2/BiVO₄, whereas the absorption enhancement of BP-1/BiVO₄ is close to Figure R11. Therefore, the absorbance different between Figure R3 and Figure R11 can be ascribed to ultrathin film that causes an optical interference, and the proportion of BP in BP/BiVO₄ that determines the absorption enhancement. The absorption enhancement across the whole spectrum

agrees well with the color change, where the BP-2/BiVO₄ displays darker yellow color (Figure R24). Notes that the proportion of BP in BP/BiVO₄ photoanode used in our manuscript is very small, therefore, the optical enhancement is relatively weak.

Another point regarding the absorber effect of BP is addressed by measuring the PEC properties of BP photoelectrode. As shown in Figure R25, the BP photoelectrode presents clear cathodic photoresponses in KPi electrolyte (pH=7.1) under AM 1.5 illumination, which is consistent with the Mott-Schottky measurement result. As pointed out by Chorkendorff et al (Energy Environ. Sci., 2014, 7, 2397), the cathodic overlayer on photoanode is dependent on the position of valance band for hole accumulation, rather than electron injection. Therefore, for the enhanced PEC performance by p-type BP, the absorber effect of BP can be reasonably excluded

Figure R23: UV-vis absorbance of pure FTO, BiVO₄ and BP/BiVO₄ with different amounts photoanodes (This Figure is for reviewers only).

Figure R24: Photographs of BP-1/BiVO₄ (left) or BP-2/BiVO₄ (right) photoanodes (This Figure is for reviewers only).

Figure R25: Reproduced J-V curves of BP photoelectrode in KPi electrolyte (pH=7.1) under AM 1.5 illumination (This Figure is for reviewers only).

Reviewer #2: Dear editor and author

The three reviewers have put forward constructive comments, which is very beneficial to the technical improvement of this article. After careful revision by the author, all questions have been answered more rigorously, and the article has become more mature and conducive to publication. So I'd recommend this article to accept after addressing the below minor question: please show me the value of C_{bulk} and C_{trap} (especially C_{trap}) to see whether charge storage is positively correlated with the fitted surface capacitance.

Our Response: We thank for your highly positive comments and many suggestions to improve our paper. In the revised manuscript, we added the fitting values of double layer capacitances, C_{bulk} and C_{trap} . At 0.6 V vs RHE bias, the bulk capacitances (C_{bulk}) for all photoanodes are almost same, which can be ascribed to the redox process of V^{4+}/V^{5+} (Energy Environ. Sci., 2017, 10, 1517-1529). The trapping capacitances (C_{trap}) for pure photoanode can be related to its localized surface states (J. Am. Chem. Soc., 2012, 134, 4294–4302), while a surface layer that modifies surface state will become a dominated role (J. Am. Chem. Soc., 2012, 134, 16693–16700). As a result, the C_{trap} are significantly increased from several to twenty times as a function of surface layers. The calculated value for NiOOH/BP/BiVO₄ is 567.5 $\mu\text{F cm}^{-2}$, which is closed to its charge storage capability (1.016 mC cm^{-2}) at 0.6 V vs RHE. The results were provided in our revised manuscript.

Table R1. Fitting results of Nyquist plots. (This Figure is included as Table S1 in the revised manuscript).

Samples	R1 (Ω)	R2 (Ω)	R3 (Ω)	C_{bulk} ($\mu\text{F cm}^{-2}$)	C_{trap} ($\mu\text{F cm}^{-2}$)
BiVO ₄	11.6	3200		35.2	29.7
BP/BiVO ₄	12.4	1084		32.9	328.9
NiOOH/FeOOH/BiVO ₄	10.3	204	1412	30.2	132
NiOOH/BP/BiVO ₄	10.2	162	1416	32.2	567.5

Reviewer #3: My concerns have been covered and I am happy to recommend publication.

Our Response: We thank for your previously constructive suggestion and the recommendation of publication

REVIEWERS' COMMENTS:

Reviewer #1 (Remarks to the Author):

The authors have made good efforts to address the comments from previous reviews, and now the manuscript should be publishable in this journal.

Reviewer #2 (Remarks to the Author):

My concerns have been covered and I am happy to recommend publication.

Response to the Comments on Nature Communications

Manuscript ID: NCOMMS-18-36462B

Dear Reviewers:

We would like to thank the reviewers for careful reading and helpful comments. We revised the manuscript thoroughly according to the comments. The added items are highlighted in red in the main manuscript and the supplementary information. Following changes were made and listed below:

Reviewer(s)' Comments to Author:

Reviewer #1: The authors have made good efforts to address the comments from previous reviews, and now the manuscript should be publishable in this journal.

Our Response: We thank for your highly positive comments and many suggestions to improve our paper.

Reviewer #2: My concerns have been covered and I am happy to recommend publication.

Our Response: We thank for your highly positive comments and many suggestions to improve our paper.